# Beyond Affinity: A Benchmark of 1D, 2D, and 3D Methods Reveals Critical Trade-offs in Structure-Based Drug Design

**Kangyu Zheng**                                                           *zhengk5@rpi.edu*
*Department of Computer Science*
*Rensselaer Polytechnic Institute*

**Kai Zhang**                                                              *kaz321@lehigh.edu*
*Department of Computer Science and Engineering*
*Lehigh University*

**Jiale Tan**                                                             *jialetan@usc.edu*
*Department of Computer Science*
*University of Southern California*

**Xuehan Chen**                                                           *Xuc321@lehigh.edu*
*Department of Computer Science and Engineering*
*Lehigh University*

**Yingzhou Lu**                                                           *lyz66@stanford.edu*
*Stanford Medicine Department of Pathology*
*Stanford University*

**Zaixi Zhang**                                                           *zz8680@princeton.edu*
*Princeton University*

**Lichao Sun**                                                            *lis221@lehigh.edu*
*Department of Computer Science and Engineering*
*Lehigh University*

**Marinka Zitnik**                                                        *marinka@hms.harvard.edu*
*Harvard Medical School*

**Tianfan Fu**                                                            *futianfan@nju.edu.cn*
*State Key Laboratory for Novel Software Technology at Nanjing University*
*Nanjing University*

**Zhiding Liang**                                                         *zliang@cse.cuhk.edu.hk*
*Department of Computer Science and Engineering*
*The Chinese University of Hong Kong*

**Reviewed on OpenReview:** *https://openreview.net/forum?id=gaTwx1rzCw*

## Abstract

Currently, the field of structure-based drug design is dominated by three main types of algorithms: search-based algorithms, deep generative models, and reinforcement learning. While existing works have typically focused on comparing models within a single algorithmic category, cross-algorithm comparisons remain scarce. In this paper, to fill the gap, we establish a benchmark to evaluate the performance of fifteen models across these different algorithmic foundations by assessing the pharmaceutical properties of the generated molecules and their docking affinities and poses with specified target proteins. We highlight the unique advantages of each algorithmic approach and offer recommendations for the design of fu-

ture SBDD models. We emphasize that 1D/2D ligand-centric drug design methods can be used in SBDD by treating the docking function as a black-box oracle, which is typically neglected. Our evaluation reveals distinct patterns across model categories. 3D structure-based models excel in binding affinities but show inconsistencies in chemical validity and pose quality. 1D models demonstrate reliable performance in standard molecular metrics but rarely achieve optimal binding affinities. 2D models offer balanced performance, maintaining high chemical validity while achieving moderate binding scores. Through detailed analysis across multiple protein targets, we identify key improvement areas for each model category, providing insights for researchers to combine strengths of different approaches while addressing their limitations. All the code that are used for benchmarking is available in `https://github.com/zkysfls/2025-sbdd-benchmark`

# 1 Introduction

Novel types of safe and effective drugs are needed to meet the medical needs of billions worldwide and improve the quality of human life. The process of discovering a new drug candidate and developing it into an approved drug for clinical use is known as *drug discovery* (Sinha & Vohora, 2018). This complex process is fundamental to the development of new therapies that can manage, cure, or alleviate the symptoms of various health conditions.

Structure-based drug design (SBDD) (Bohacek et al., 1996) is a core strategy that accelerates this process by using the three-dimensional (3D) structures of disease-related proteins to develop drug candidates. This approach is grounded in the "lock and key" model (Tripathi & Bankaitis, 2017), where molecules that bind more effectively to a target protein are more likely to modulate its function, a principle validated by numerous experimental studies (Honarparvar et al., 2014; Blundell, 1996; Lu et al., 2022).

Currently, three main algorithmic approaches dominate the drug design field (Brown et al., 2019; Gao et al., 2022; Du et al., 2022): search-based algorithms like genetic algorithms (GA) (Jensen, 2019; Spiegel & Durrant, 2020; Tripp & Hernández-Lobato, 2023; Fu et al., 2022a), deep generative models such as variational autoencoder (VAE) (Gómez-Bombarelli et al., 2018) and autoregressive models (Luo et al., 2021; Peng et al., 2022; Zhang et al., 2023), and reinforcement learning (RL) models (Olivecrona et al., 2017; Zhou et al., 2019). While these models, particularly those using 3D protein representations (Zhang et al., 2023; Luo et al., 2021; Fu et al., 2022a; Peng et al., 2022), are considered state-of-the-art for generating valid and diverse molecules, a clear comparative understanding across these different algorithmic foundations is lacking. Existing benchmarks often focus on comparing models within the same category (typically deep generative models) and prioritize molecular properties over the crucial evaluation of protein-ligand interactions. (Du et al., 2022; Brown et al., 2019; Gao et al., 2022).

A preliminary study first highlighted that 1D/2D models can achieve competitive performance against 3D methods in structure-based drug design (Zheng et al., 2024). Our work substantially extends these initial findings by introducing a more rigorous and multi-faceted evaluation framework. We curates a comprehensive benchmark that encompasses fifteen models spanning all three algorithmic approaches. We assess model performance through multiple dimensions: traditional heuristic molecular property oracles, docking scores, and pose evaluations - all critical metrics for understanding the quality of protein-ligand interactions in the context of drug discovery. Our analysis reveals a notable dichotomy in 3D models' performance: while they excel in docking score optimization, achieving consistently superior binding affinity predictions, they show comparable or sometimes inferior performance in pose quality assessments and other heuristic property evaluations. These findings highlight the need for future structure-based models to better balance docking score optimization with other crucial molecular properties and structural validity metrics.

# 2 Related Works

Significant progress has been made in benchmarking drug design models (Brown et al., 2019; Tripp et al., 2021; Huang et al., 2021; Gao et al., 2022; Polykovskiy et al., 2020; Harris et al., 2023; Zheng et al.,

Table 1: Representative structure-based drug design methods, categorized based on the molecular assembly strategies and the optimization algorithms. Columns are various molecular assembly strategies while rows are different optimization algorithms.

|  | 1D SMILES/SELFIES | 2D Molecular Graph | 3D Structure-based |
|---|---|---|---|
| Genetic Algorithm (GA) | SMILES-GA (Yoshikawa et al., 2018) | Graph GA (Jensen, 2019) | - |
| Hill Climbing | SMILES-LSTM-HC (Brown et al., 2019) | MIMOSA (Fu et al., 2021) | - |
| Reinforcement Learning (RL) | REINVENT (Olivecrona et al., 2017) | MolDQN (Zhou et al., 2019) | - |
| Gradient Ascent (GRAD) | Pasithea (Shen et al., 2021) | DST (Fu et al., 2022b) | - |
| Generative Models | SMILES/SELFIES-VAE-BO (Gómez-Bombarelli et al., 2018) | JT-VAE (Jin et al., 2019) | 3DSBDD (Luo et al., 2021), Pocket2mol (Peng et al., 2022), PocketFlow (Jiang et al., 2024), ResGen (Zhang et al., 2023), TargetDiff (Guan et al., 2023) |

Table 2: Top 1 docking score for each target. Targets in CrossDocking are marked in red, and targets not in CrossDocking are in blue.

| Model | 6GL8 | 1UWH | 7OTE | 1KKQ | 5WFD | 7W7C | 8JJL | 7D42 | 7S1S | 6AZV |
|---|---|---|---|---|---|---|---|---|---|---|
| Pocket2Mol | **-11.56** | **-14.56** | **-15.72** | **-14.18** | -11.30 | -13.76 | **-13.27** | -12.76 | **-12.90** | **-12.36** |
| PocketFlow | -9.42 | -11.04 | -10.27 | -12.47 | -10.30 | -13.52 | -11.88 | **-12.79** | -10.16 | -11.83 |
| ResGen | − | -9.71 | -7.14 | -9.81 | − | -7.88 | − | -8.94 | -11.77 | -9.71 |
| 3DSBDD | -8.61 | -12.67 | -10.52 | -13.36 | -11.28 | -11.29 | -11.67 | -11.12 | -10.19 | -10.13 |
| DST | -8.69 | -11.09 | -11.41 | -10.92 | -10.13 | -12.14 | -12.20 | -11.87 | -11.54 | -10.31 |
| graph-GA | -8.47 | -11.19 | -11.15 | -10.61 | -9.66 | -11.85 | -10.72 | -11.03 | -10.47 | -9.85 |
| JT-VAE | -10.26 | -12.38 | -12.29 | -12.25 | **-11.65** | -12.45 | -11.91 | **-12.79** | -11.53 | -10.60 |
| MIMOSA | -8.64 | -11.13 | -11.49 | -11.00 | -9.91 | -11.72 | -11.85 | -11.72 | -11.96 | -10.27 |
| MolDQN | -6.63 | -7.38 | -7.49 | -7.35 | -7.79 | -7.75 | -8.94 | -7.30 | -8.42 | -8.04 |
| Pasithea | -9.25 | -11.47 | -11.56 | -10.45 | -10.54 | -12.00 | -11.87 | -11.76 | -11.35 | -10.24 |
| REINVENT | -9.06 | -11.13 | -12.03 | -11.19 | -10.18 | -11.88 | -11.63 | -12.23 | -11.32 | -10.66 |
| SMILES-GA | -8.83 | -10.74 | -11.18 | -10.47 | -9.72 | -11.74 | -11.29 | -11.93 | -11.05 | -10.46 |
| SMILES-LSTM-HC | -9.77 | -11.50 | -12.35 | -12.40 | -11.21 | **-13.93** | -11.41 | -12.84 | -12.02 | -10.61 |
| SMILES-VAE | -9.35 | -11.95 | -13.06 | -10.91 | -10.01 | -12.11 | -11.95 | -12.01 | -12.14 | -10.42 |
| TargetDiff | − | -7.30 | − | -11.53 | -10.07 | -9.27 | -8.91 | -6.10 | − | − |

2024). Benchmarks like Guacamol, Molecular Sets (MOSES), Practical Molecule Optimization (PMO), and POSECHECK have been crucial for evaluating algorithms on molecular properties and protein-ligand interactions. More recently, CBGBench (Lin et al., 2024) introduced a comprehensive framework for evaluating various 3D SBDD models across multiple generative tasks (de novo, linker design, scaffold hopping).

Our work builds upon these efforts but occupies a distinct niche by providing the first large-scale, cross-paradigm benchmark specifically comparing 1D, 2D, and 3D methods side-by-side for the de novo structure-based generation task under unified evaluation conditions. Crucially, we extend beyond standard metrics by incorporating detailed pose quality assessments using both PoseBuster (Buttenschoen et al., 2024) and PoseCheck (Harris et al., 2023), alongside geometric consistency checks (redocking RMSD analysis), enabling a focused diagnosis of the critical "affinity-validity trade-off". We benchmark fifteen models across these paradigms, grouping them into three categories based on their molecular representation to analyze their distinct strengths and weaknesses in the context of SBDD.

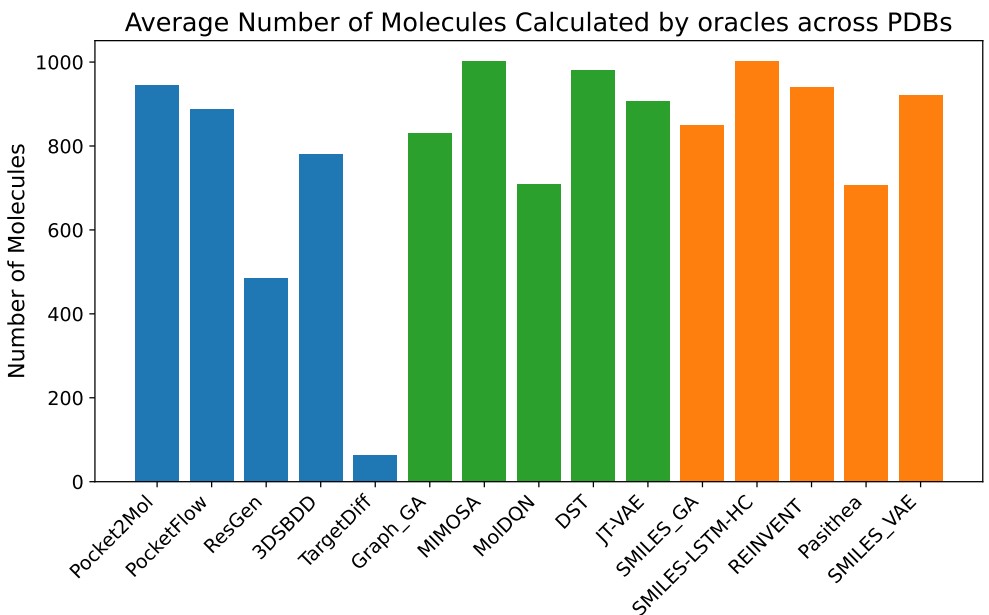

Figure 1: The bar chart of average generated molecules that are calculated by our selected oracles for each model across all target proteins under given time. 1D methods are colored orange, blue is used to indicate 3D methods, and green represents 2D methods.

**1D Molecule Design Methods** 1D molecule design methods use Simplified Molecular-Input Line-Entry System (SMILES) (Weininger, 1988) or SELF-referencing Embedded Strings (SELFIES) (Lo et al., 2023) strings as the representation of molecules. Most 1D methods produce molecule strings in an autoregressive manner. In this paper, we discuss several methods that were developed to produce molecule strings, either SMILES or SELFIES strings, including REINVENT (Olivecrona et al., 2017), SMILES and SELFIES VAE (Gómez-Bombarelli et al., 2018), SMILES GA (Yoshikawa et al., 2018), SMILES-LSTM-HC (Brown et al., 2019), and Pasithea (Shen et al., 2021). Although SELFIES string has the advantage of enforcing chemical validity rules compared to SMILES, through thorough empirical studies, Gao et al. (2022) showed that SELFIES string-based methods do not demonstrate superiority over SMILES string-based ones.

**2D Molecule Design Methods** Compared to 1D molecule design methods, representing molecules using 2D molecular graphs is a more sophisticated approach. molecular 2D representation, graphs are used to depict molecules, where edges represent chemical bonds and nodes represent atoms. There are two main strategies for constructing these graphs: atom-based and fragment-based. Atom-based methods operate on one atom or bond at a time, searching the entire chemical space. On the other hand, fragment-based methods summarize common molecular fragments and operate on one fragment at a time, which can be more efficient. In this paper, we discuss several methods belonging to this category: MolDQN (Zhou et al., 2019), which uses an atom-based strategy, and Graph GA (Jiang et al., 2024), Multi-constraint Molecule Sampling (MIMOSA) (Fu et al., 2021), Differentiable Scaffolding Tree (DST) (Fu et al., 2022b), JT-VAE (Jin et al., 2019) which use fragment-based.

**3D Molecule Design Methods** Both 1D and 2D molecule design methods are ligand-centric, focusing primarily on designing the molecule itself. In structure-based drug design, as pointed out in Huang et al. (2021), these models take the docking function as a black box, which inputs a molecule and outputs the binding affinity score. However, these models fail to incorporate target protein structure information and consequently suffer from high computational time (to find binding pose). In contrast, 3D structure-based drug design methods take the three-dimensional geometry of the target protein as input and directly generate pocket-aware molecules in the pocket of target protein. In this paper, we cover five cutting-edge structure-based drug design methods: TargetDiff (Guan et al., 2023), PocketFlow (Jiang et al., 2024), 3DSBDD (Luo et al., 2021), Pocket2mol (Peng et al., 2022), and ResGen (Zhang et al., 2023).

A critical challenge for these 3D methods is ensuring SE(3) equivariance, guaranteeing that rotations and translations of the input protein-ligand system lead to correspondingly transformed outputs, maintaining physical consistency. Different approaches tackle this differently. ResGen, Pocket2Mol, and PocketFlow utilize specialized equivariant components like Geometric Vector Perceptrons (Jing et al., 2021) or equivariant attention processing scalar and vector features. In contrast, 3DSBDD relies on invariant inter-atomic distances for GNN message passing. TargetDiff specifically combines an equivariant GNN with Center of Mass normalization for overall invariance.

## 3 Models

In this paper, the models we select for evaluation are based on one or a combination of the following algorithms. For ease of comparison, we categorize all the methods based on optimization algorithm and molecular assembly strategy in Table 1.

**Genetic Algorithm (GA)**: Inspired by natural selection, genetic algorithm is a combinatorial optimization method that evolves solutions to problems over many generations. Specifically, in each generation, GA will perform crossover and mutation over a set of candidates to produce a pool of offspring and keep the top-k offspring for the next generation, imitating the natural selection process. In our evaluation, we choose three GA models: SMILES GA (Yoshikawa et al., 2018) that performs GA over SMILES string-based space, Graph GA (Jiang et al., 2024) that searches over atom- and fragment-level by designing their crossover and mutation rules on graph matching.

**Variational Auto-Encoder (VAE) and Diffusion**: The aim of variational autoencoder is to generate new data that is similar to training data. In the molecule generation area, VAE learns a bidirectional map between molecule space and continuous latent space and optimizes the latent space. VAE itself generated diverse molecules that are learned from the training set. After training VAE, Bayesian optimization (BO) is used to navigate latent space efficiently, identify desirable molecules, and conduct molecule optimization. Diffusion models represent another class of generative models. They operate based on a two-stage process: a fixed 'forward' or 'diffusion' process that gradually adds noise (e.g., Gaussian noise for coordinates, categorical noise for atom types) to the data over a series of time steps, eventually transforming it into a simple prior distribution. A learned 'reverse' or 'denoising' process then trains a neural network to reverse this, starting from noise and iteratively removing the noise at each time step to generate a sample that resembles the original data distribution. In our evaluation, we select three VAE-based models and one diffusion-based model: SMILES-VAE-BO (Gómez-Bombarelli et al., 2018) uses SMILES string as the input to the VAE model, and SELFIES-VAE-BO uses the same algorithm but uses SELFIES string as the molecular representation. JT-VAE (Jin et al., 2019) instead use a tree-structured scaffold over chemical substructures. TargetDiff (Guan et al., 2023) design a target-aware diffusion model that could generate molecules under given protein atoms.

**Auto-regressive**: An auto-regressive model is a type of statistical model that is based on the idea that past values in the series can be used to predict future values. In molecule generation, an auto-regressive model would typically take the generated atom sequence as input and predict which atom would be the next. In our evaluation, we choose seven auto-regressive models: PocketFlow (Jiang et al., 2024) is autoregressive flow-based generative models. 3DSBDD (Luo et al., 2021) based on conventional Markov Chain Monte Carlo (MCMC) algorithms and Pocket2mol (Peng et al., 2022) choose graph neural networks (GNN) as the backbone. Inspired by Pocket2mol, ResGen (Zhang et al., 2023) used a hierarchical autoregression, which consists of a global autoregression for learning protein-ligand interactions and atomic component autoregression for learning each atom's topology and geometry distributions.

**Hill Climbing (HC)**: Hill Climbing (HC) is an optimization algorithm that belongs to the family of local search techniques (Selman & Gomes, 2006). It is used to find the best solution to a problem among a set of possible solutions. In molecular design, Hill Climbing would tune the generative model with the reference of generated high-scored molecules. In our evaluation, we adopt two HC models: SMILES-LSTM-HC (Brown et al., 2019) uses an LSTM model to generate molecules and uses the HC technique to fine-tune it. MultI-constraint MOlecule SAmpling (MIMOSA) (Fu et al., 2021) uses a graph neural network instead and incorporates it with HC.

**Gradient Ascent (GRAD)**: Similar to gradient descent, gradient ascent also estimates the gradient direction but chooses the maximum direction. In molecular design, the GRAD method is often used in molecular property function to optimize molecular generation. In our evaluation, we choose two GRAD-based models: Pasithea (Shen et al., 2021) uses SELFIES as input and applies GRAD on an MLP-based molecular property prediction model. Differentiable Scaffolding Tree (DST) (Fu et al., 2022b) uses differentiable molecular graph as input and uses a graph neural network to estimate objective and the corresponding gradient.

**Reinforcement Learning (RL)**: In molecular generation context, a reinforcement learning model would take a partially-generated molecule (either sequence or molecular graph) as state; action is how to add a token or atom to the sequence or molecular graph respectively; and reward is the property score of current molecular sequence. In our evaluation, we test on two RL-based models: REINVENT (Olivecrona et al., 2017) is a policy-gradient method that uses RNN to generate molecules and MolDQN (Zhou et al., 2019) uses a deep Q-network to generate molecular graph.

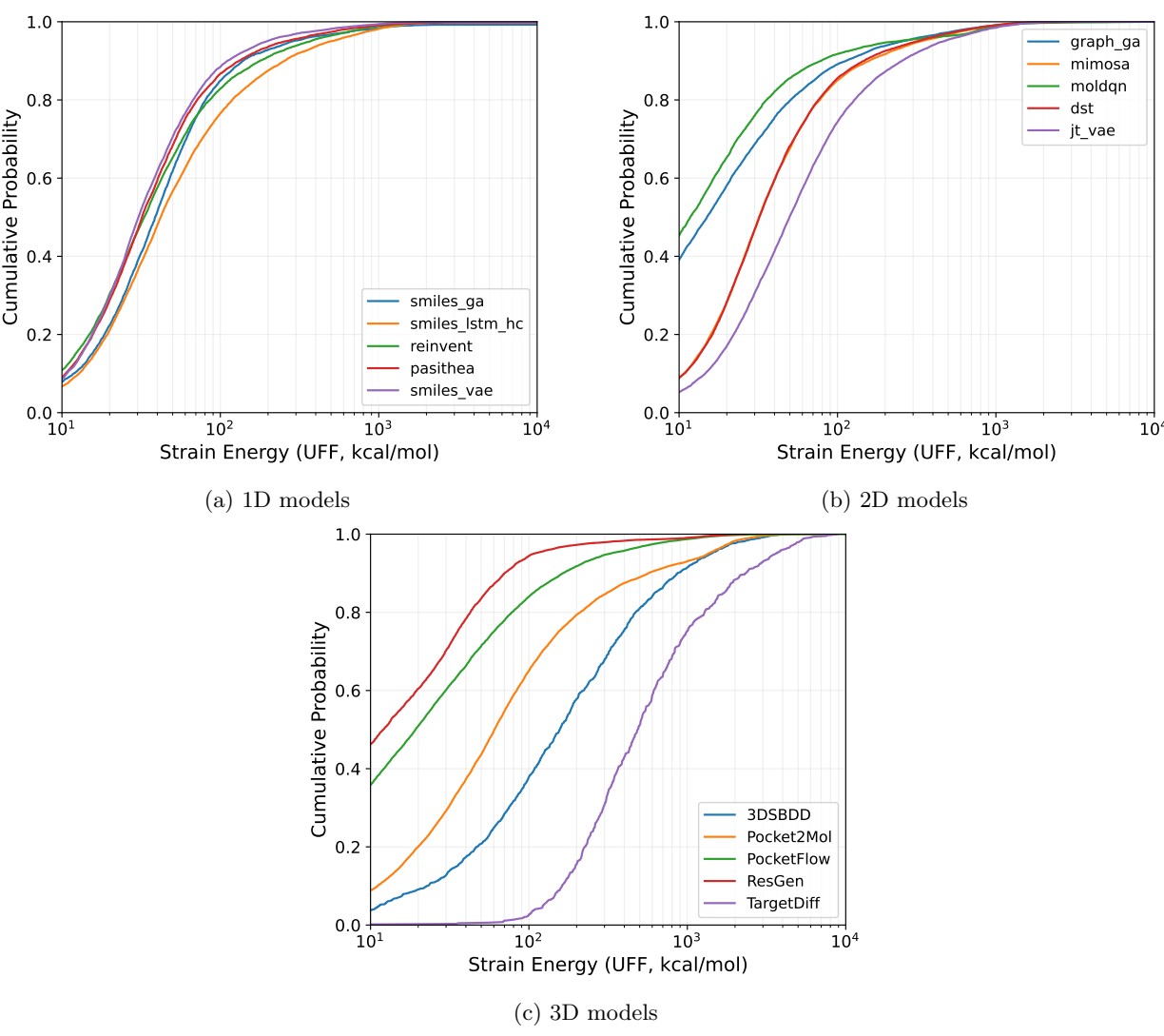

(a) 1D models

(b) 2D models

(c) 3D models

Figure 2: The cumulative density function (CDF) of strain energy of each model

# 4    Experiments

In this section, we demonstrate the experimental results. We start with the description of experimental setup. Then, we present and analyze the experimental results, including protein-ligand bindings and pose, pharmaceutical properties of generated molecules (e.g., drug-likeness and synthetic accessibility), and other qualities of generated molecules (e.g., diversity, validity).

## 4.1    Experiment Setup

### 4.1.1    Oracle

In drug discovery, we need to evaluate the pharmaceutical properties of the generated molecules, such as binding affinity to certain target proteins, drug-likeness, synthetic accessibility, solubility, etc. These property evaluators are also known as *oracle*. In this section, we introduce the oracle we chose to evaluate these models. The oracle functions are sourced from either the Therapeutics Data Commons (TDC) (Huang et al., 2022; 2021)[1] or established Python packages.

**(1) Docking Score**: Molecular docking is a measurement of free energy exchange between a ligand and a target protein during the binding process. A lower docking score means the ligand would have a higher potential to pose higher bioactivity with a given target. Compared with other heuristic oracles, such as QED (quantitative estimate of drug-likeness), and LogP (Octanol-water partition coefficient), docking reflects the binding affinities between drug molecule and target (Graff et al., 2021). Our experiments use AutoDock Vina (Eberhardt et al., 2021) python package to calculate the docking score for each generated molecules. We selected ten representative and diverse target proteins, with five sourced from CrossDock (Francoeur et al., 2020) and five from other datasets. The CrossDock-derived PDB IDs are 6GL8, 1UWH, 7OTE, 1KKQ, and 5WFD, while the remaining structures are 7WC7, 8JJL, 7D42, 6AZV, and 7S1S. These crystallography structures are across different fields, including virology, immunology, and oncology (Huang et al., 2022; 2021; Chang et al., 2019). They cover various kinds of diseases such as chronic myelogenous leukemia, tuberculosis, SARS-COVID-2, etc. They represent a breadth of functionality, from viral replication mechanisms to cellular signaling pathways and immune responses.

**(2) Pose**: Pose evaluation is another crucial oracle in structure-based drug discovery. It primarily focuses on measuring physical and chemical validity aspects of docking, such as chemical consistency, geometric plausibility, and energy-based checks. These assessments help eliminate physically implausible poses that might score well in Vina but are chemically or geometrically invalid. Additionally, it evaluates intermolecular interactions between the ligand and protein, such as steric clashes, to ensure that the predicted poses are not only energetically favorable but also physically realistic in the protein environment. In our experiments, we employed two pose evaluation packages: PoseBuster (Buttenschoen et al., 2024) and PoseCheck (Harris et al., 2023). PoseBuster contains 19 True/False-style metrics spanning chemical validity and consistency, intramolecular validity, and intermolecular validity. PoseCheck provides numerical values for clashes and strain energy for each docking pose generated by Vina.

**(3) Heuristic Oracles**: Although heuristic oracles are considered to be "trivial" and too easily optimized, we still incorporate some of them into our evaluation metrics for comprehensive analysis. In our experiments, we utilize Quantitative Estimate of Drug-likeness (QED), SA, and LogP as our heuristic oracles. QED evaluates a molecule's drug-likeness on a scale from 0 to 1, where 0 indicates minimal drug-likeness and 1 signifies maximum drug-likeness, aligning closely with the physicochemical properties of successful drugs. SA, or Synthetic Accessibility, assesses the ease of synthesizing a molecule, with scores ranging from 1 to 10; a lower score suggests easier synthesis. LogP measures a compound's preference for a lipophilic (oil-like) phase over a hydrophilic (water-like) phase, essentially indicating its solubility in water, where the optimal range depends on the type of drug. But mostly the value should be between 0 and 5.

**(4) Molecule Generation Oracles**: While docking score oracles and heuristic oracles focus on evaluating individual molecules, molecule generation oracles assess the quality of all generated molecules as a whole. In our experiments, we choose three metrics to evaluate the generated molecules of each model: diversity,

---

[1]https://tdcommons.ai/functions/oracles/

validity, and uniqueness. Diversity is measured by the average pairwise Tanimoto distance between the Morgan fingerprints (Benhenda, 2017). Validity is determined by checking atoms' valency and the consistency of bonds in aromatic rings using RDKit's molecular structure parser (Polykovskiy et al., 2020). Uniqueness is measured by the frequency at which a model generates duplicated molecules, with lower values indicating more frequent duplicates (Polykovskiy et al., 2020).

### 4.1.2 Model Setup

Inspired by Liu et al. (2024), we setup our experiment as follow: For each model, we aim to generate 1,000 molecules for each given target protein. In case there are models fail to sample sufficient enough molecules at one run, we will rerun models that failed at most three times with different seed. We do not retrain each model and instead we directly use the checkpoints they provided. We use the receptor information from Liu et al. (2024) when setting target. After we get enough molecules for each model, we apply the four oracles mentioned above to every molecule and collect the results. None of the tested models have prior knowledge of these oracle functions.

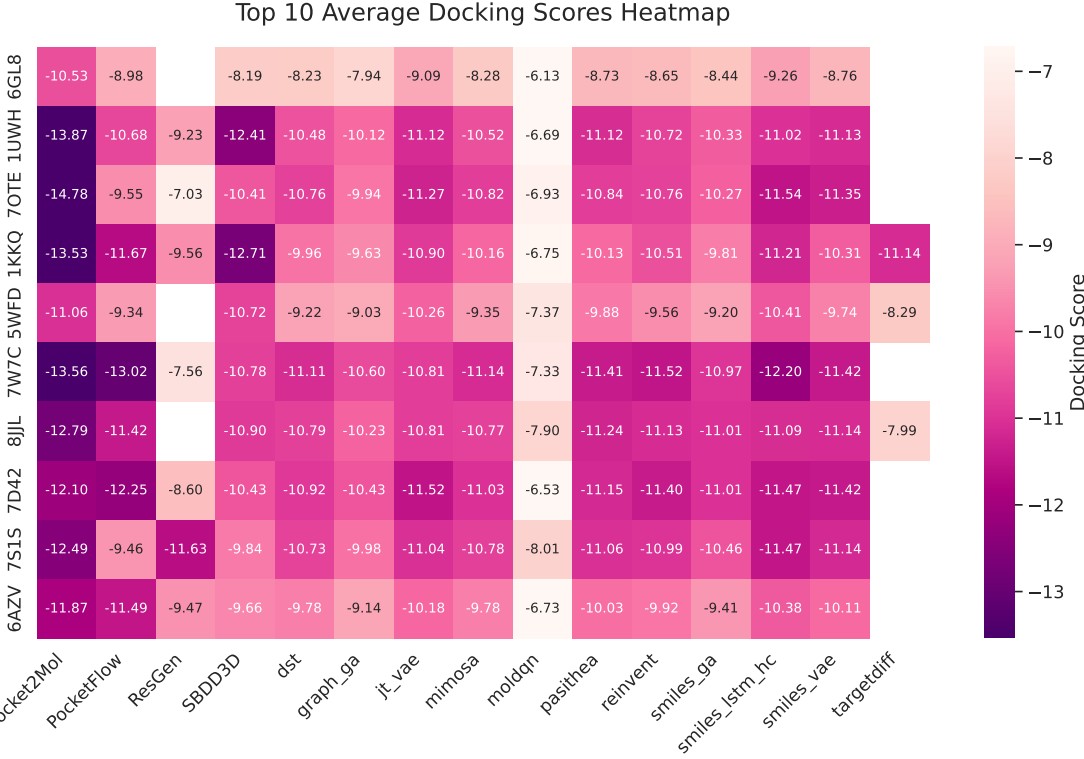

Figure 3: The heatmap based on the average of each model's Top-10 docking score for each target protein.

### 4.2 Experiment Results

In this section, we present our experimental results. First, we analyze the generation performance of each model, including the number of molecules successfully generated and stored, as well as generation speed. We then evaluate the oracle results for the generated molecules. Finally, we discuss our observations regarding model performance based on these analyses.

### 4.2.1 Generation Performance and Efficiency

In our analysis of model generation capabilities, we examined both generation speed and the total number of successfully stored molecules. As specified in the model setup section, each model was tasked with generating

1,000 molecules per run. We measured generation speed in molecules per minute, calculated by dividing the total generation time by 1,000. The speed results are presented in Table 3.

Table 3: Comparison of model running times (molecules/minute) across different targets. 'error' means we fail to record the time due to model error. Targets in CrossDocking are marked in red, and targets not in CrossDocking are in blue.

| Model | 6GL8 | 1UWH | 7OTE | 1KKQ | 5WFD | 7W7C | 8JJL | 7D42 | 7S1S | 6AZV |
|---|---|---|---|---|---|---|---|---|---|---|
| SMILES-GA | 6.23 | 4.44 | 5.71 | 5.16 | 8.63 | 6.00 | 4.58 | 5.61 | 6.25 | 6.60 |
| SMILES-LSTM-HC | 1.20 | 0.97 | 0.94 | 1.01 | 1.11 | 1.10 | 1.43 | 1.31 | 0.95 | 0.96 |
| REINVENT | 0.54 | 0.50 | 0.54 | 0.54 | 0.52 | 0.50 | 0.50 | 0.52 | 0.52 | 0.59 |
| Pasithea | 1.88 | error | error | 1.79 | 1.87 | error | error | error | 1.78 | 1.83 |
| SMILES-VAE | 0.75 | 0.68 | 0.71 | error | 0.75 | 0.76 | 0.70 | 0.78 | 0.72 | 0.73 |
| graph GA | 3.21 | 2.92 | error | 3.20 | 3.33 | 3.47 | error | 3.40 | 3.28 | 3.26 |
| MIMOSA | 1.55 | 1.35 | 1.46 | 1.51 | 1.56 | 1.58 | 1.40 | 1.60 | 1.48 | 1.52 |
| MolDQN | | | | | error | | | | | |
| DST | 0.48 | 0.43 | 0.48 | 0.48 | 0.48 | 0.48 | 0.43 | 0.48 | 0.48 | 0.48 |
| JT-VAE | 1.40 | 1.07 | 1.29 | 1.36 | 1.39 | error | 1.21 | 1.47 | 1.42 | 1.35 |
| TargetDiff | 1.18 | 0.65 | 0.66 | 0.64 | 0.22 | 0.49 | 0.45 | 0.79 | 0.67 | 0.28 |
| 3DSBDD | 1.85 | 3.02 | 5.95 | 2.26 | 4.67 | 2.82 | 3.50 | 1.68 | 3.06 | 2.94 |
| Pocket2mol | **66.66** | 40 | 40 | 37.03 | **90.91** | 37.52 | **43.47** | 45.45 | 43.47 | 43.47 |
| PocketFlow | 33.33 | 33.33 | 27.78 | 28.57 | 15.15 | 16.39 | 18.86 | 21.74 | 23.26 | 19.23 |
| ResGen | error | **225.35** | **359.71** | **125** | error | **333.3** | error | **172.41** | **125** | **111.11** |

Several models encountered execution issues with specific targets, resulting in 'error' designations. These failures occurred for various reasons: immediate execution errors (as with Pasithea), infinite loop conditions (as observed with MolDQN), and initialization failures due to unsuitable molecular configurations (as seen with ResGen). Notably we observe that ResGen and TargetDiff failed to generate molecules on serval targets. For ResGen, we hypothesize the issue comes from its sensitivity to the input pocket definition coupled with its residue-level multiscale protein representation. If the specific receptor configurations used in our benchmark deviate significantly from ResGen's training assumptions, the initial step of placing the first atom may fail. For TargetDiff, we suspect the low yield of valid molecules to two main factors: Inaccurate prediction of the number of atoms to generate, which is done prior to generation based on estimated pocket size relative to training data bins; And failures in the post-hoc bond reconstruction step when the diffused atom cloud geometry is unrealistic. It is important to note that the absence of recorded time does not necessarily indicate a complete failure to generate and store molecules.

Among successfully executing models, we first examined 1D and 2D models from the PMO benchmark (Gao et al., 2022), which underwent identical processing. Most of these models achieved average speeds of 1-3 molecules per minute. SMILES-GA demonstrated superior performance with speeds of 6-7 molecules per minute, followed by Graph-GA at 3-4 molecules per minute. The relatively modest speeds may be attributed to the computational overhead of PMO's internal oracle calculations, particularly for complex molecular structures.

In the 3D model category, ResGen achieved the highest average speed of approximately 300 molecules per minute. However, it failed to initiate generation for several targets (6GL8, 5WFD, 1UWH) given our receptor specifications. Pocket2mol maintained the second-highest performance at 40-60 molecules per minute, while PocketFlow operated at approximately 30 molecules per minute.

Regarding the number of stored molecules (Table 9), several models consistently approached or reached the 1,000-molecule target. SMILES-LSTM-HC, REINVENT, MIMOSA, and PocketFlow demonstrated notably stable performance, consistently generating and storing the full 1,000 molecules across most targets. However, some models showed significant variability. TargetDiff particularly struggled, storing fewer than 100 molecules for most targets, with its best performance being 557 molecules for 1KKQ. JT-VAE showed inconsistent performance, notably generating only 200 molecules for receptor 7WC7 despite strong performance

with other targets. 3DSBDD displayed varying success rates, ranging from 222 to 1,000 stored molecules across different targets.

### 4.2.2 Binding Affinities and Poses

| Method | Median Strain |
| --- | --- |
| graph-GA | 14.99 |
| MIMOSA | 32.01 |
| MolDQN | 11.75 |
| DST | 32.06 |
| JT-VAE | 49.70 |
| SMILES-GA | 38.66 |
| SMILES-LSTM-HC | 41.95 |
| REINVENT | 33.11 |
| Pasithea | 31.99 |
| SMILES-VAE | 30.35 |
| 3DSBDD | 157.11 |
| Pocket2Mol | 60.55 |
| PocketFlow | 19.21 |
| ResGen | 12.05 |
| TargetDiff | 487.72 |

Table 4: Median strain energies across different methods.

We first take a look about the docking score performance by the Top-1 (Table 2), Top-10 average (Table 10 and Figure 3) and Top-100 average (Table 11). Overall, the docking scores generally range from -14 to -5, with more negative values indicating better binding affinity. Pocket2Mol consistently demonstrates superior performance across all three tables, with score ranging from between -10 and -15. In contrast, MolDQN shows the weakest performance, with scores consistently around -5 to -7. The remaining models typically score between -8 and -12, with 3D models generally achieving the best scores across most targets.

**1D models**: The 1D models (SMILES-GA, SMILES-LSTM-HC, REINVENT, SMILES-VAE) show consistent performance across targets. In particular, SMILES-VAE and REINVENT demonstrate strong results with scores around -10 to -11 in the Top-10 average. SMILES-LSTM-HC exhibits slightly better performance for CrossDocking targets compared to non-CrossDocking ones. SMILES-GA maintains performance comparable to other 1D models but shows higher variability across different targets.

**2D models**: The 2D models exhibit more diverse performance patterns compared to 1D models. DST and MIMOSA achieve scores comparable to 1D models, demonstrating competitive performance. Graph-GA maintains consistency but performs slightly below the top 1D models. JT-VAE scores well but shows notable variance across targets. MolDQN consistently underperforms relative to other 2D approaches.

**3D models**: Among 3D models, Pocket2Mol emerges as the clear leader, consistently achieving top scores. PocketFlow also performs strongly but shows higher variance between targets. ResGen delivers moderate but consistent performance. 3DSBDD displays the most variable performance among 3D models, achieving some excellent scores but with significant variance.

To statistically validate these observations, we performed paired t-tests comparing the average top-10 docking scores across shared targets for representative models: SMILES-LSTM-HC for 1D model, JT-VAE for 2D model and Pocket2mol for 3D model. We found that Pocket2mol achieved significantly better scores than both SMILES-LSTM-HC (1D) (absolute mean difference = 1.65 kcal/mol, $t = -5.86$, $p = 0.0002$) and JT-VAE (2D) (absolute mean difference = 1.96 kcal/mol, $t = -6.58$, $p = 0.0001$). We also observe that JT-VAE performs slightly better than SMILES-LSTM-HC (absolute mean difference = 0.31 kcal/mol, $t = 2.34$, $p = 0.044$). These results statistically confirm the superior binding affinity prediction of the 3D models.

The above analysis demonstrates that while 3D models, particularly Pocket2Mol, can achieve the best overall scores, each category of models has its distinct performance characteristics and trade-offs.

We next examine the pose-related results, beginning with the Posebusters evaluation results presented in Table 27. Each value in the table represents the pass rate for all molecules generated by the corresponding model. For Chemical Validity and Consistency tests (including mol pred loaded, mol cond loaded, sanitization, inchi convertible, and all atoms connected), 2D models demonstrate consistently perfect performance, while 3D models show notable variations. Specifically, 3DSBDD and Pocket2Mol exhibit lower pass rates in all-atom connectivity and inchi convertibility tests. SMILES-based models show lower but consistent performance in this category, particularly in inchi convertible and all-atom connectivity metrics. Regarding Intramolecular Validity tests (covering bond lengths/angles, internal steric clash, aromatic ring flatness, double bond flatness, and internal energy), most models perform well except in the Internal Energy category. Here, significant variations are observed, with some models (3DSBDD, Pocket2Mol) showing very low pass rates while others (DST, graph-GA) achieve excellent results. For Intermolecular Validity tests (encompassing protein-ligand maximum distance, minimum distances to protein/cofactors/waters, and volume overlap with protein/cofactors/waters), nearly all models achieve perfect pass rates, with TargetDiff being the notable exception, showing poor performance in minimum distance to protein metrics. In summary, while most models excel at intermolecular validity checks, the key differences emerge in chemical validity and intramolecular metrics. 2D models maintain the most consistent performance across all categories, whereas 3D models and SMILES-based approaches display greater variability, particularly in chemical validity assessments.

We also use PoseCheck to calculate the strain energy and the steric clash of the generated poses. A clash occurs when the pairwise distance between protein and ligand atoms falls below the sum of their van der Waals radii, which is physically implausible (Ramachandran et al., 2011; Buonfiglio et al., 2015). Strain energy represents the internal energy within a ligand during binding, with lower energy typically being more favorable (Perola & Charifson, 2004).

First, we examine strain energy results in Figure 2 and Table 4, which reveal distinct patterns across model dimensions. 3D models show wide variation, with ResGen performing best (median: 12.05 kcal/mol) and TargetDiff worst (median: 487.72 kcal/mol). 2D models generally perform better, led by MolDQN (11.75 kcal/mol) and graph-GA (14.99 kcal/mol). 1D SMILES-based models show consistent performance, with SMILES-VAE (30.35 kcal/mol) outperforming others in this category.

Turning to clash results shown in Figure 5, Figure 6, Figure 7 and Table 28, we observe significant variation among 3D models. ResGen demonstrates the best performance with consistently low clash counts (median 2-4) across most receptors, while Pocket2Mol and TargetDiff exhibit high variability. In contrast, 1D and 2D models show stable performance with most models having median clash counts no higher than 10. Interestingly, receptor 8JJL consistently proves challenging for models across all categories.

To further evaluate the quality of the poses generated by the 3D models, we performed a redocking experiment. For each molecule, we calculated the Root-Mean-Square Deviation (RMSD) between the model's original generated pose and the pose found by AutoDock Vina. This metric serves as a sanity check to determine if the generated geometries are consistent with a physics-based scoring function. A lower RMSD indicates a better alignment. The results are show in Figure 4 and Table 5. We observe that ResGen produces poses most consistent with Vina, with a median RMSD of 2.21 Å and 42.7% of its poses falling under the 2.0 Å success threshold. PocketFlow also performs reasonably well. In contrast, models like Pocket2Mol and 3DSBDD, which often achieved the highest binding affinities, perform poorly on this metrics.

| Model | Median RMSD (Å) | Std. Dev. (Å) | % Poses < 2.0 Å |
|---|---|---|---|
| 3DSBDD | 5.35 | 2.38 | 2.97% |
| Pocket2Mol | 4.03 | 1.40 | 5.40% |
| PocketFlow | 2.75 | 1.61 | 29.69% |
| ResGen | **2.21** | 1.47 | **42.73%** |
| TargetDiff | 4.63 | 1.09 | 0.16% |

Table 5: RMSD statistics comparison of all 3D models

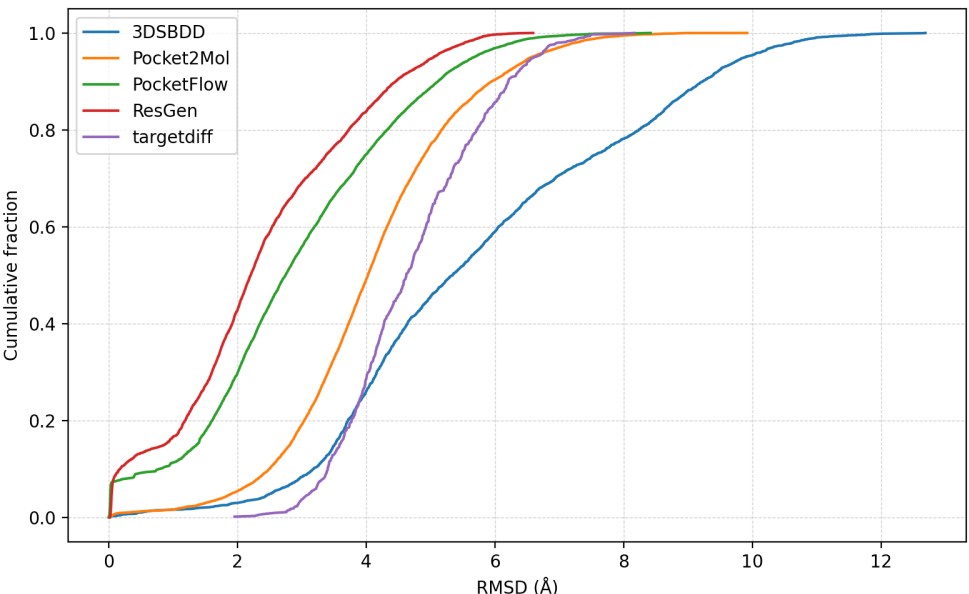

Figure 4: The cumulative density function (CDF) of RMSD for all 3D models

In summary, while 3D models often achieve better docking scores, they do not necessarily produce better protein-ligand geometric compatibility.

### 4.2.3 Pharmaceutical Properties

In this section, we report and analyze the pharmaceutical properties of the generated molecules.

**LogP**: Overall, nearly all the models tested produced the majority of their molecules within the 0 to 3 range. For example, SMILES-LSTM-HC demonstrates remarkable consistency, maintaining a LogP score of 3.39 ± 0.60 across all targets. Several models, including Pasithea, SMILES-VAE, MIMOSA, and DST, show stable performance with LogP values between 2.40 and 2.60, indicating good potential for drug-like properties. However, there is significant variability among other models. Pocket2Mol and PocketFlow exhibit wide fluctuations in their LogP scores, ranging from -0.05 to 5.24, suggesting inconsistent control over molecular hydrophobicity. MolDQN consistently generates highly hydrophilic molecules, with LogP scores consistently below -1.0 across all targets. 3DSBDD shows the most extreme variability, with scores ranging from -11.36 to 3.73, and notably large standard deviations.

**QED**:Based on the results in table 17, these models demonstrate varying levels of performance across different targets. The majority of models, including SMILES-VAE, MIMOSA, and DST, consistently achieve high QED scores between 0.90-0.91, indicating their robust ability to generate drug-like molecules. A second tier of models, including SMILES-GA, SMILES-LSTM-HC, and JT-VAE, maintains reliable performance with scores ranging from 0.86 to 0.88. The graph-GA model shows moderate consistency with scores around 0.84 across all targets. Notably, there are significant performance disparities among certain models. MolDQN demonstrates considerably lower performance, with QED scores consistently falling between 0.43-0.49 across all targets. PocketFlow and 3DSBDD show variable performance, with scores fluctuating between 0.60 and 0.80, suggesting less stability in generating drug-like molecules. The performance patterns remain relatively consistent whether evaluating targets in CrossDocking or those not included in CrossDocking, indicating that the models' capabilities are generally independent of the target selection.

**SA**: Overall, most of the models generate molecules with scores between 1 to 3. Several models maintain consistent performance with relatively low variance: SMILES-LSTM-HC (1.93 ± 0.13), MIMOSA (1.95 ± 0.12), and DST (1.95 ± 0.14) demonstrate stable synthetic accessibility scores across all targets. However, 3DSBDD shows significantly higher scores with substantial variance, ranging from 2.89 ± 0.73 to 6.10 ± 0.96.

MolDQN consistently generates molecules with higher SA scores (2.83-3.28) and notable standard deviations (0.50-0.55). The data indicates that while most models can generate synthetically accessible compounds, there are significant variations in their consistency and the complexity of the proposed synthetic routes.

To provide a more modern assessment of synthetic complexity beyond the heuristic SA score, we also calculated the SCScore (Coley et al., 2018) and RAscore (Thakkar et al., 2021) for the generated molecules. SCScore predicts the complexity of a molecule based on reaction precedence data, with lower scores indicating potentially easier synthesis (typically ranging from 1 to 5). RAscore predicts the retrosynthetic accessibility of a molecule based on whether an AI synthesis planning tool can find a route, with higher scores (near 1) indicating a route was found and lower scores (near 0) indicating a route was not found. The results of SCScore are in table 21, 22, 23 and the results of RAscore are in table 24, 25, 26.

We first observe that most 1D and 2D models consistently generated molecules with lower SCscores (often < 2.5). In contrast, some 3D models, particularly 3DSBDD, tended to produce molecules with higher average SCscores (e.g., > 3.5 on several targets), indicating potentially greater synthetic complexity. These findings generally reinforce the trends observed with the SA score, consistent with the strong correlation (Spearman $\rho = 0.88$) between the two metrics in our dataset. RAscore on the other hand lacks of differentiation. The values are consistently very high (close to 1.0) across almost all models and targets and therefore show minimal variation. Its correlation with SA Score (Spearman $\rho = 0.68$) and SCScore (Spearman $\rho = 0.69$) was also weaker than the SA vs. SCscore correlation.

Furthermore, we report the numerical values of top-$K$ docking, QED, SA, and LogP scores for all the methods across different target proteins in the Appendix A.

### 4.2.4 Molecule Generation Quality

| Model | Diversity | Uniqueness | Validity |
|---|---|---|---|
| SMILES-GA | 0.90 | 0.60 | 1.00 |
| SMILES-LSTM-HC | 0.88 | 0.17 | 1.00 |
| REINVENT | 0.89 | 0.70 | 1.00 |
| Pasithea | 0.91 | 0.28 | 1.00 |
| SMILES-VAE | 0.88 | 0.54 | 1.00 |
| graph-GA | 0.92 | 0.71 | 1.00 |
| MIMOSA | 0.88 | 0.09 | 1.00 |
| MolDQN | 0.93 | 0.71 | 1.00 |
| DST | 0.88 | 0.10 | 1.00 |
| JT-VAE | 0.89 | 0.77 | 1.00 |
| TargetDiff | 0.89 | 1.00 | 1.00 |
| Pocket2Mol | 0.84 | 1.00 | 1.00 |
| PocketFlow | 0.89 | 0.84 | 1.00 |
| ResGen | 0.85 | 0.97 | 1.00 |
| 3DSBDD | 0.89 | 0.80 | 0.80 |

Table 6: average Molecule Generation Metrics across all target

We record the results of Molecule Generation Oracles in Table 6 and below are the breakdown analysis.

In the diversity oracle, all models demonstrates strong capabilities with scores consistently above 0.84. MolDQN achieves the highest diversity at 0.93, followed closely by graph-GA (0.92) and Pasithea (0.91). This high-performing cluster suggests that these models are particularly adept at generating structurally varied molecules. The remaining models maintain robust diversity scores between 0.84 and 0.90, indicating generally strong performance in exploring chemical space.

In the validity oracle, nearly all models shows exceptional performance with fourteen out of fifteen models achieving perfect scores of 1. This indicates these models consistently generate chemically valid molecular structures. 3DSBDD stands as the sole exception with a validity score of 0.80, suggesting some room for improvement in ensuring chemical validity of its generated structures.

Table 7: Model ranking summary

| Model | clash | docking | generation | heuristic | posebusters | strain | overall rank |
|---|---|---|---|---|---|---|---|
| PocketFlow | 5 | 4 | 4 | 2 | 6 | 4 | 1 |
| graph-GA | 3 | 12 | 3 | 7 | 1 | 3 | 2 |
| SMILES-VAE | 7 | 5 | 11 | 1 | 9 | 5 | 3 |
| MolDQN | 1 | 15 | 2 | 11 | 4 | 1 | 4 |
| ResGen | 2 | 14 | 9 | 8 | 5 | 2 | 5 |
| Pasithea | 8 | 7 | 7 | 5 | 12 | 6 | 6 |
| REINVENT | 4 | 8 | 7 | 7 | 10 | 9 | 7 |
| JT-VAE | 12 | 3 | 5 | 6 | 3 | 12 | 8 |
| MIMOSA | 9 | 9 | 13 | 4 | 2 | 7 | 9 |
| DST | 10 | 10 | 12 | 3 | 2 | 8 | 10 |
| SMILES-LSTM-HC | 13 | 2 | 10 | 2 | 8 | 11 | 11 |
| Pocket2Mol | 14 | 1 | 8 | 3 | 11 | 13 | 12 |
| SMILES-GA | 6 | 11 | 6 | 9 | 13 | 10 | 13 |
| 3DSBDD | 11 | 6 | 14 | 10 | 14 | 14 | 14 |
| TargetDiff | 15 | 13 | 1 | 12 | 7 | 15 | 15 |

Uniqueness scores reveal the most significant variations among the three metrics. TargetDiff and Pocket2Mol achieve perfect uniqueness scores of 1.00, while ResGen (0.97) and PocketFlow (0.84) also demonstrate strong performance. However, there is a notable performance gap, with some models showing substantially lower uniqueness scores. MIMOSA (0.09), DST (0.10), and SMILES-LSTM-HC (0.17) generate relatively high proportions of duplicate structures. This variance in uniqueness scores suggests that while models can generate valid and diverse molecules, ensuring uniqueness remains a challenging aspect of molecular generation.

This comprehensive analysis indicates that while the field has largely solved the challenge of generating valid molecular structures and can consistently produce diverse molecule sets, the ability to generate unique molecules varies significantly across different approaches.

## 5 Discussion

### 5.1 Overall Ranking

While the primary goal of this benchmark is to highlight the trade-offs between different approaches, providing a summarized view can offer additional guidance. To this end, we calculated an overall ranking based on the models' performance across the diverse metrics evaluated. We first ranked the models within each of the six major categories: clash count, docking score, generation quality (diversity, uniqueness, validity), heuristic properties (QED, SA, LogP), PoseBusters overall pass rates, and strain energy. These six category ranks were then averaged for each model, applying equal weight to each category, to determine the final overall rank presented in Table 7. It is crucial, however, to interpret this ranking with caution. The assumption of equal weighting across all metric categories is subjective. Different drug discovery campaigns may prioritize certain aspects much more heavily. Therefore, while this table provides a useful summary, the detailed results across individual metrics remain essential for selecting the most appropriate model for a specific scientific goal.

### 5.2 Metric Limitations

We acknowledge certain scope limitations in this benchmark. Due to computational and time constraints inherent in evaluating fifteen distinct generative models, we focused on AutoDock Vina as the primary docking oracle for cross-paradigm comparison and utilized a set of ten diverse protein targets. Integrating alternative state-of-the-art docking or affinity prediction methods, such as DiffDock (Corso et al., 2023)

and AF3 (Abramson et al., 2024), or substantially expanding the target set remain important directions for future work but were infeasible for this initial comprehensive comparison.

### 5.3 Analysis and Future Approaches

Our benchmark experiments reveal a nuanced performance landscape across 1D, 2D, and 3D algorithmic families. This subsection provides a deeper analysis of these results, explaining the underlying reasons for the observed performance trade-offs and offering insights into the strategic application of these different approaches in drug discovery.

Intuitively, 3D models should have a decisive advantage, as they explicitly use the 3D coordinates of the target's binding pocket as input. This allows them to learn the geometric and chemical constraints imposed by the pocket's shape, directly optimizing for steric and electrostatic complementarity. Our results confirm this, showing that 3D models consistently generate molecules with superior binding affinity. However, we also observe inconsistent chemical validity and poor pose quality. There are several possible factors that can explain this "affinity-validity trade-off":

- **Sequential Error Accumulation**: Many 3D models are autoregressive, building molecules atom-by-atom. This sequential process can suffer from "exposure bias," where small geometric prediction errors in early steps accumulate, resulting in final structures with high internal strain or steric clashes, even if the overall shape fits the pocket.

- **Primacy of Coordinates over Chemistry**: The intense focus on optimizing 3D coordinates can come at the expense of enforcing fundamental, discrete rules of chemistry. This can lead to geometrically plausible but chemically unfavorable arrangements, such as strained or uncommon ring structures. Some models successfully mitigate this by explicitly incorporating chemical knowledge, like valence rules, into the generation process to improve validity.

- **Practical Hurdles**: 3D models often face practical constraints that can limit their use. They can be computationally intensive and may lack robustness, sometimes failing to generate molecules for novel protein targets without specific configurations—an issue we observed in our own experiments.

On the contrary, 1D and 2D models have better performance on chemical validity may because they are using more chemical "language" such as SMILES string or fragments and scaffolds. However these methods treat target as a "black-box oracle", generating a molecule first and only then checking its fit. This indirect optimization is inefficient for discovering high-affinity binders.

Given the complementary strengths and weaknesses of each approach, the most promising direction for future research lies in the development of hybrid models. To test this hypothesis directly, we evaluated a recent hybrid model, TamGen (Wu et al., 2024), which uses 3D protein context to guide 1D molecule generation. The results, summarized in Table 8, provide a crucial and nuanced perspective. Contrary to a simple "hybrid is best" narrative, we found that TamGen did not resolve the core trade-offs observed in our benchmark. Its binding affinity was lower than our representative 1D, 2D, and 3D models. More significantly, it produced molecules with the highest median strain energy and the lowest chemical validity of the group. These results suggest that this particular hybrid implementation struggles with enforcing correct chemical geometries, a challenge we had primarily associated with pure 3D models.

This finding is highly instructive. It demonstrates that simply combining model modalities is not a guaranteed solution. The specific architectural choices for how 3D structural information is integrated with 1D/2D chemical grammars are critically important. This underscores the complexity of the SBDD challenge and refines our conclusion: the future lies not just in hybrid models, but in hybrid architectures that explicitly and robustly enforce chemical validity throughout the generation process.

## 6 Conclusion

Currently, the landscape of structure-based drug design models is vast, featuring various algorithmic backbones, yet comparative analyses across them are scarce. In this study, we design experiments to evaluate

Table 8: A case study comparing the hybrid model (TamGen) against representative models. The results are chosen from the best of each selected models achieved

| Metric | 1D Model (SMILES-LSTM-HC) | 2D Model (JT-VAE) | 3D Model (Pocket2Mol) | Hybrid Model (TamGen) |
|---|---|---|---|---|
| Top-10 Docking Score (Affinity) | -12.20 | -11.52 | **-14.78** | -10.66 |
| Median Strain Energy (kcal/mol) | **41.95** | 49.70 | 60.55 | 76.26 |
| Top-10 generation metrics (Diversity/Uniqueness/Validity) | 0.88/0.17/1.0 | 0.89/0.77/1.0 | 0.84/1.0/1.0 | 1.0/0.10/0.42 |
| Top-10 heuristic (SA/LogP/QED) | 1.66/4.64/0.93 | 1.60/3.87/0.93 | 1.05/4.38/0.93 | 1.99/3.38/0.90 |

the quality of molecules generated by each model. Our experiments extend beyond conventional heuristic oracles related to molecular properties, also examining the affinity and poses between molecules and selected target proteins. Our findings indicate that different algorithmic approaches exhibit distinct strengths and limitations. 3D models, particularly Pocket2Mol, demonstrate superior performance in generating molecules with strong binding affinities, as evidenced by docking scores. However, they show more variability in chemical validity checks and pose evaluations. 1D SMILES-based models maintain consistent performance across most metrics but rarely achieve the highest binding affinities. 2D graph-based models offer the most balanced performance, showing strong consistency across chemical validity, pose quality, and moderate binding affinity scores.

These results suggest that while significant progress has been made in structure-based drug design, there remains room for improvement in developing models that can simultaneously optimize binding affinity while maintaining high chemical validity and pose quality. Future development of structure-based models should focus on bridging these gaps, potentially by incorporating mechanisms that better balance the trade-offs between binding affinity optimization and other crucial molecular properties. This comprehensive evaluation provides valuable insights for researchers working to advance the field of computational drug discovery.

**Acknowledgement.** Tianfan Fu is supported by Young Scientists Fund (C Class) of the National Natural Science Foundation of China (Grant No. 62506154) and Nanjing University International Collaboration Initiative.

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

## A  Appendix

Table 9: Number of successfully generated and stored molecules by model. Targets in CrossDocking are marked in red, and targets not in CrossDocking are in blue.

| Model | 6GL8 | 1UWH | 7OTE | 1KKQ | 5WFD | 7WC7 | 8JJL | 7D42 | 6AZV | 7S1S |
|---|---|---|---|---|---|---|---|---|---|---|
| SMILES-GA | 832 | 800 | 819 | 892 | 580 | 791 | 652 | 662 | 629 | 558 |
| SMILES-LSTM-HC | 1000 | 1000 | 1000 | 1000 | 1000 | 1000 | 1000 | 1000 | 1000 | 1000 |
| REINVENT | 1000 | 1000 | 1000 | 1000 | 1000 | 1000 | 1000 | 1000 | 1000 | 1000 |
| Pasithea | 1000 | 800 | 900 | 1000 | 1000 | 800 | 900 | 800 | 1000 | 1000 |
| SMILES-VAE | 1000 | 1000 | 1000 | 600 | 1000 | 1000 | 1000 | 1000 | 1000 | 1000 |
| graph GA | 1000 | 1000 | 400 | 1000 | 1000 | 1000 | 500 | 1000 | 1000 | 1000 |
| MIMOSA | 1000 | 1000 | 1000 | 1000 | 1000 | 1000 | 1000 | 1000 | 1000 | 1000 |
| MolDQN | 700 | 690 | 700 | 800 | 700 | 700 | 700 | 700 | 700 | 696 |
| DST | 1000 | 900 | 1000 | 1000 | 1000 | 1000 | 900 | 1000 | 1000 | 1000 |
| JT-VAE | 1000 | 1000 | 1000 | 1000 | 1000 | 200 | 1000 | 1000 | 1000 | 1000 |
| TargetDiff | 0 | 1 | 0 | 557 | 70 | 4 | 10 | 1 | 0 | 0 |
| 3DSBDD | 898 | 708 | 669 | 668 | 769 | 1000 | 1000 | 1000 | 222 | 673 |
| Pocket2mol | 1000 | 968 | 1000 | 790 | 1000 | 1000 | 860 | 884 | 868 | 1000 |
| PocketFlow | 1000 | 1000 | 1000 | 1000 | 1000 | 1000 | 1000 | 1000 | 1000 | 1000 |
| ResGen | 0 | 676 | 290 | 785 | 0 | 259 | 0 | 805 | 1000 | 1000 |

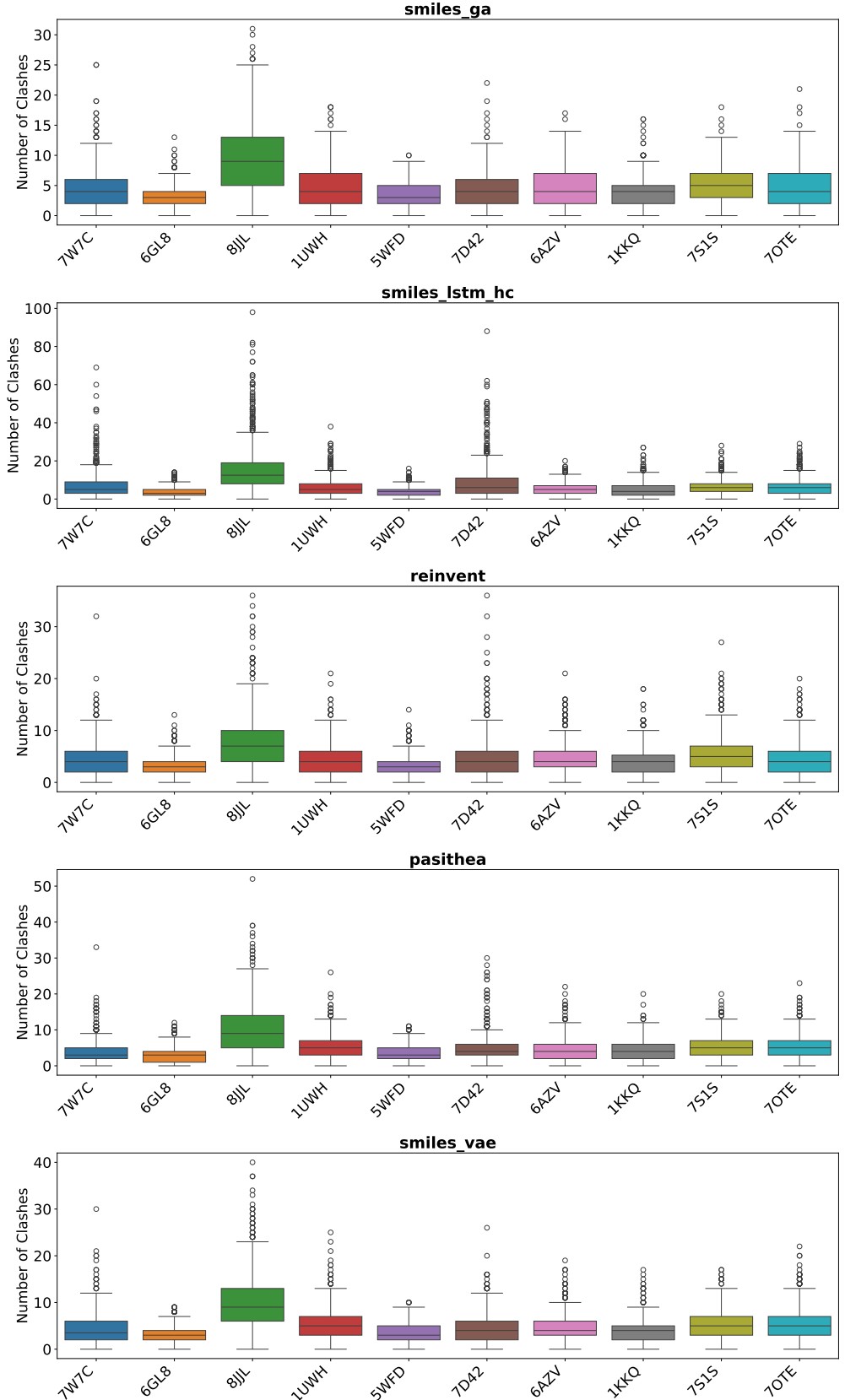

Figure 5: The clashes box plot for 1D models

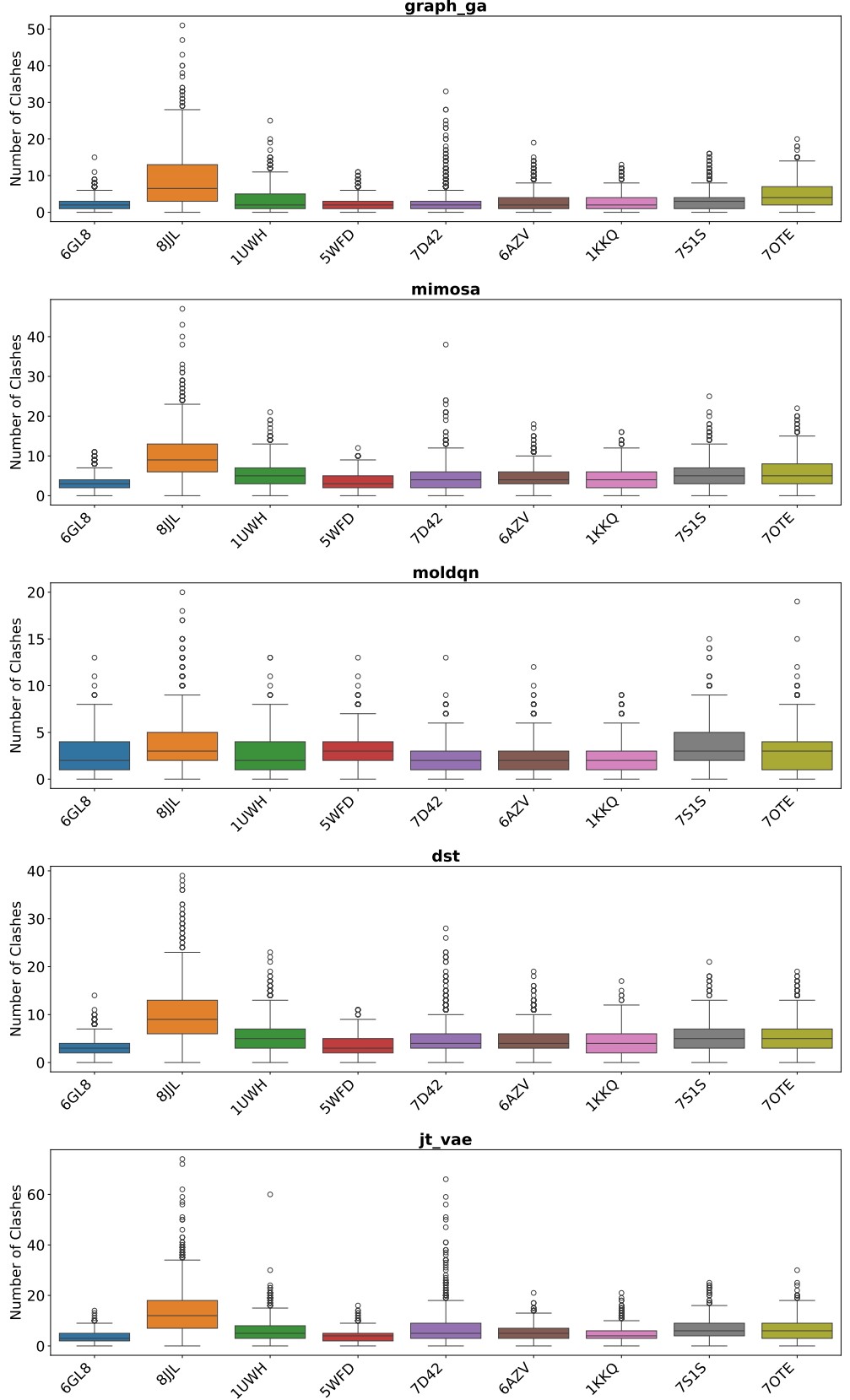

Figure 6: The clashes box plot for 2D models

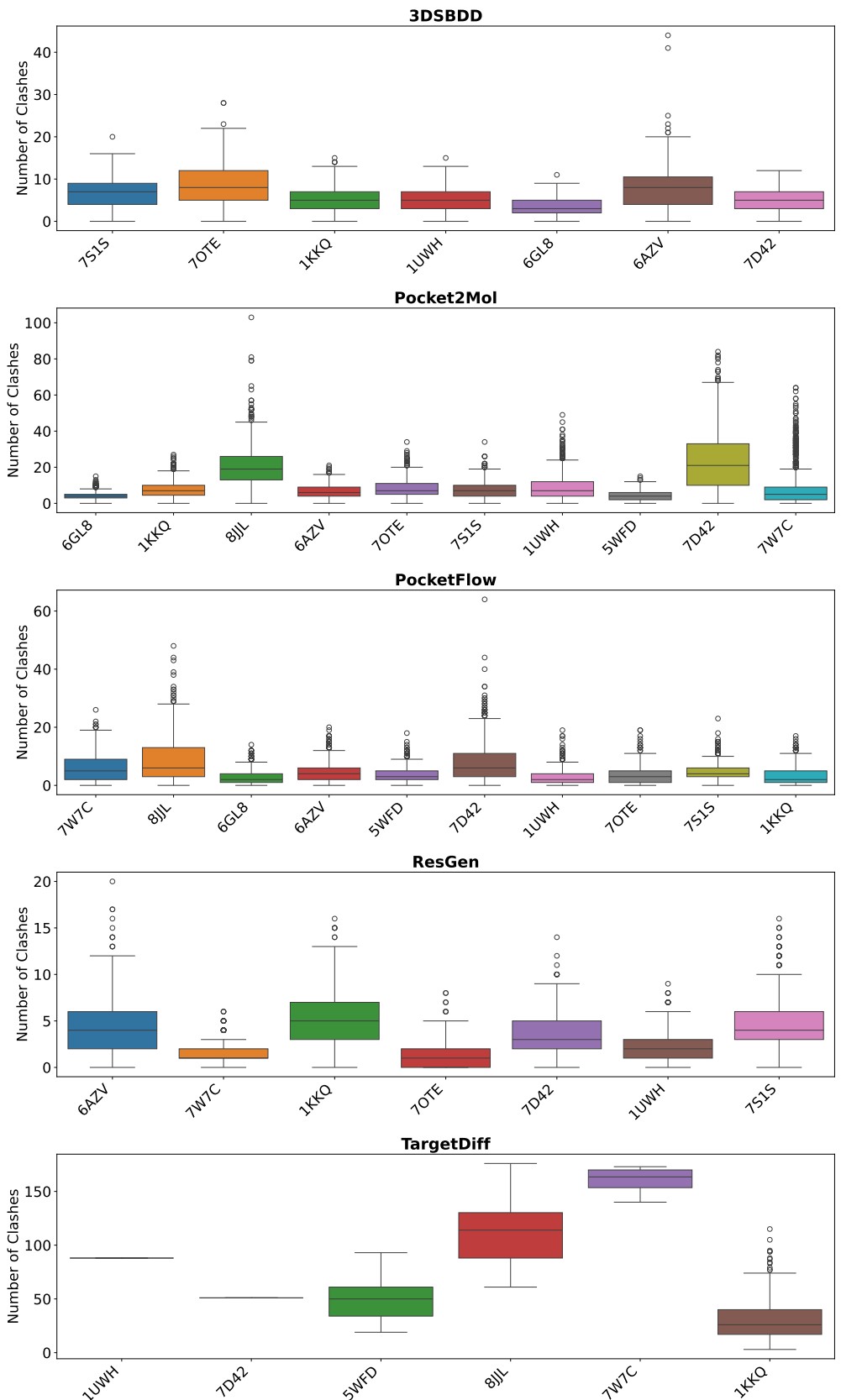

Figure 7: The clashes box plot for 3D models

| Model | 6GL8 | 1UWH | 7OTE | 1KKQ | 5WFD |
|---|---|---|---|---|---|
| Pocket2Mol | **-10.53** ± 0.40 | **-13.87** ± 0.34 | **-14.78** ± 0.44 | **-13.53** ± 0.30 | **-11.06** ± 0.15 |
| PocketFlow | -8.98 ± 0.22 | -10.68 ± 0.23 | -9.55 ± 0.36 | -11.67 ± 0.38 | -9.34 ± 0.35 |
| ResGen | – | -9.23 ± 0.20 | -7.03 ± 0.06 | -9.56 ± 0.18 | – |
| 3DSBDD | -8.19 ± 0.22 | -12.41 ± 0.18 | -10.41 ± 0.09 | -12.71 ± 0.30 | -10.72 ± 0.22 |
| DST | -8.23 ± 0.15 | -10.48 ± 0.16 | -10.76 ± 0.21 | -9.96 ± 0.32 | -9.22 ± 0.28 |
| graph-GA | -7.94 ± 0.18 | -10.12 ± 0.37 | -9.94 ± 0.36 | -9.63 ± 0.32 | -9.03 ± 0.18 |
| JT-VAE | -9.09 ± 0.42 | -11.12 ± 0.41 | -11.27 ± 0.34 | -10.90 ± 0.47 | -10.26 ± 0.42 |
| MIMOSA | -8.28 ± 0.13 | -10.52 ± 0.22 | -10.82 ± 0.25 | -10.16 ± 0.25 | -9.35 ± 0.19 |
| MolDQN | -6.13 ± 0.17 | -6.69 ± 0.25 | -6.93 ± 0.20 | -6.75 ± 0.21 | -7.37 ± 0.14 |
| Pasithea | -8.73 ± 0.24 | -11.12 ± 0.27 | -10.84 ± 0.26 | -10.13 ± 0.18 | -9.88 ± 0.33 |
| REINVENT | -8.65 ± 0.24 | -10.72 ± 0.28 | -10.76 ± 0.44 | -10.51 ± 0.28 | -9.56 ± 0.24 |
| SMILES-GA | -8.44 ± 0.23 | -10.33 ± 0.27 | -10.27 ± 0.39 | -9.81 ± 0.38 | -9.20 ± 0.30 |
| SMILES-LSTM-HC | -9.26 ± 0.29 | -11.02 ± 0.25 | -11.54 ± 0.40 | -11.21 ± 0.51 | -10.41 ± 0.42 |
| SMILES-VAE | -8.76 ± 0.25 | -11.13 ± 0.37 | -11.35 ± 0.61 | -10.31 ± 0.31 | -9.74 ± 0.16 |
| TargetDiff | – | – | – | -11.14 ± 0.26 | -8.29 ± 0.69 |

(a) Targets in CrossDocking.

| Model | 7W7C | 8JJL | 7D42 | 7S1S | 6AZV |
|---|---|---|---|---|---|
| Pocket2Mol | **-13.56** ± 0.11 | **-12.79** ± 0.33 | -12.10 ± 0.33 | **-12.49** ± 0.26 | **-11.87** ± 0.26 |
| PocketFlow | -13.02 ± 0.23 | -11.42 ± 0.28 | **-12.25** ± 0.30 | -9.46 ± 0.34 | -11.49 ± 0.15 |
| ResGen | -7.56 ± 0.18 | – | -8.60 ± 0.20 | -11.63 ± 0.08 | -9.47 ± 0.13 |
| 3DSBDD | -10.78 ± 0.28 | -10.90 ± 0.35 | -10.43 ± 0.29 | -9.84 ± 0.21 | -9.66 ± 0.32 |
| DST | -11.11 ± 0.37 | -10.79 ± 0.48 | -10.92 ± 0.32 | -10.73 ± 0.30 | -9.78 ± 0.16 |
| graph-GA | -10.60 ± 0.39 | -10.23 ± 0.19 | -10.43 ± 0.20 | -9.98 ± 0.16 | -9.14 ± 0.21 |
| JT-VAE | -10.81 ± 0.54 | -10.81 ± 0.39 | -11.52 ± 0.41 | -11.04 ± 0.16 | -10.18 ± 0.14 |
| MIMOSA | -11.14 ± 0.21 | -10.77 ± 0.36 | -11.03 ± 0.29 | -10.78 ± 0.33 | -9.78 ± 0.18 |
| MolDQN | -7.33 ± 0.16 | -7.90 ± 0.35 | -6.53 ± 0.26 | -8.01 ± 0.16 | -6.73 ± 0.39 |
| Pasithea | -11.41 ± 0.26 | -11.24 ± 0.32 | -11.15 ± 0.26 | -11.06 ± 0.18 | -10.03 ± 0.12 |
| REINVENT | -11.52 ± 0.19 | -11.13 ± 0.25 | -11.40 ± 0.34 | -10.99 ± 0.20 | -9.92 ± 0.30 |
| SMILES-GA | -10.97 ± 0.29 | -11.01 ± 0.18 | -11.01 ± 0.43 | -10.46 ± 0.35 | -9.41 ± 0.40 |
| SMILES-LSTM-HC | -12.20 ± 0.60 | -11.09 ± 0.18 | -11.47 ± 0.46 | -11.47 ± 0.24 | -10.38 ± 0.13 |
| SMILES-VAE | -11.42 ± 0.33 | -11.14 ± 0.42 | -11.42 ± 0.27 | -11.14 ± 0.34 | -10.11 ± 0.19 |
| TargetDiff | – | -7.99 ± 0.53 | – | – | – |

(b) Targets not in CrossDocking.

Table 10: Top 10 Vina score for each target.

| Model | 6GL8 | 1UWH | 7OTE | 1KKQ | 5WFD |
|---|---|---|---|---|---|
| Pocket2Mol | -9.90 ± 0.31 | -12.81 ± 0.48 | -13.14 ± 0.78 | -12.36 ± 0.52 | -10.45 ± 0.29 |
| PocketFlow | -7.92 ± 0.52 | -8.12 ± 1.43 | -7.74 ± 0.85 | -10.33 ± 0.60 | -8.14 ± 0.56 |
| ResGen | – | -8.06 ± 0.56 | -6.30 ± 0.48 | -8.82 ± 0.35 | – |
| 3DSBDD | -7.22 ± 0.44 | -11.52 ± 0.44 | -9.67 ± 0.38 | -11.59 ± 0.51 | -9.86 ± 0.40 |
| DST | -7.50 ± 0.28 | -9.41 ± 0.40 | -9.47 ± 0.49 | -9.00 ± 0.40 | -8.41 ± 0.31 |
| graph-GA | -6.81 ± 0.40 | -8.32 ± 0.68 | -8.25 ± 0.65 | -8.07 ± 0.59 | -7.65 ± 0.49 |
| JT-VAE | -7.93 ± 0.48 | -9.82 ± 0.51 | -9.65 ± 0.59 | -9.62 ± 0.53 | -9.00 ± 0.48 |
| MIMOSA | -7.49 ± 0.27 | -9.54 ± 0.38 | -9.49 ± 0.48 | -8.99 ± 0.40 | -8.46 ± 0.32 |
| MolDQN | -4.84 ± 0.47 | -5.46 ± 0.47 | -5.87 ± 0.40 | -5.65 ± 0.43 | -5.79 ± 0.56 |
| Pasithea | -7.93 ± 0.39 | -9.94 ± 0.51 | -9.98 ± 0.43 | -9.36 ± 0.36 | -8.92 ± 0.44 |
| REINVENT | -7.82 ± 0.36 | -9.70 ± 0.43 | -9.77 ± 0.48 | -9.40 ± 0.51 | -8.80 ± 0.36 |
| SMILES-GA | -7.37 ± 0.46 | -8.94 ± 0.66 | -8.95 ± 0.64 | -8.64 ± 0.50 | -8.12 ± 0.44 |
| SMILES-LSTM-HC | -8.29 ± 0.42 | -10.18 ± 0.41 | -10.40 ± 0.50 | -10.10 ± 0.49 | -9.43 ± 0.43 |
| SMILES-VAE | -7.97 ± 0.37 | -9.97 ± 0.50 | -10.09 ± 0.56 | -9.08 ± 0.52 | -8.84 ± 0.40 |
| TargetDiff | – | – | – | -9.84 ± 0.58 | – |

a): targets in CrossDocking

| Model | 7W7C | 8JJL | 7D42 | 7S1S | 6AZV |
|---|---|---|---|---|---|
| Pocket2Mol | -12.55 ± 0.49 | -11.63 ± 0.51 | -10.88 ± 0.56 | -11.51 ± 0.46 | -11.11 ± 0.36 |
| PocketFlow | -11.94 ± 0.47 | -10.35 ± 0.46 | -11.11 ± 0.48 | -7.82 ± 0.74 | -10.56 ± 0.40 |
| ResGen | -7.04 ± 0.24 | – | -7.71 ± 0.44 | -11.06 ± 0.31 | -8.75 ± 0.32 |
| 3DSBDD | -8.67 ± 1.14 | -9.58 ± 0.57 | -9.10 ± 0.64 | -8.95 ± 0.42 | -8.28 ± 0.64 |
| DST | -10.06 ± 0.41 | -9.74 ± 0.45 | -9.92 ± 0.41 | -9.78 ± 0.37 | -8.90 ± 0.30 |
| graph-GA | -8.89 ± 0.64 | -8.52 ± 0.61 | -8.82 ± 0.57 | -8.46 ± 0.52 | -7.88 ± 0.48 |
| JT-VAE | -8.63 ± 0.79 | -9.78 ± 0.43 | -10.30 ± 0.48 | -10.07 ± 0.35 | -9.02 ± 0.42 |
| MIMOSA | -10.08 ± 0.40 | -9.74 ± 0.45 | -9.88 ± 0.45 | -9.77 ± 0.41 | -8.85 ± 0.34 |
| MolDQN | -6.12 ± 0.43 | -6.57 ± 0.53 | -5.41 ± 0.42 | -6.38 ± 0.53 | -5.78 ± 0.38 |
| Pasithea | -10.53 ± 0.42 | -10.22 ± 0.45 | -10.31 ± 0.40 | -10.23 ± 0.42 | -9.30 ± 0.35 |
| REINVENT | -10.46 ± 0.48 | -10.26 ± 0.41 | -10.32 ± 0.48 | -10.08 ± 0.41 | -9.19 ± 0.34 |
| SMILES-GA | -9.78 ± 0.55 | -9.61 ± 0.62 | -9.39 ± 0.66 | -9.30 ± 0.50 | -8.50 ± 0.43 |
| SMILES-LSTM-HC | -11.09 ± 0.50 | -10.16 ± 0.43 | -10.62 ± 0.43 | -10.51 ± 0.44 | -9.52 ± 0.37 |
| SMILES-VAE | -10.54 ± 0.42 | -10.17 ± 0.45 | -10.34 ± 0.51 | -10.22 ± 0.44 | -9.32 ± 0.36 |
| TargetDiff | – | – | – | – | – |

b): targets not in CrossDocking

Table 11: Top 100 Average docking score for each target

Table 12: Top 1 LogP score for each target. Targets in CrossDocking are marked in red, and targets not in CrossDocking are in blue.

| Model | 6GL8 | 1UWH | 7OTE | 1KKQ | 5WFD | 7W7C | 8JJL | 7D42 | 7S1S | 6AZV |
|---|---|---|---|---|---|---|---|---|---|---|
| SMILES-GA | 3.59 | 3.59 | 3.59 | 3.59 | 3.59 | 3.59 | 3.59 | 3.59 | 3.59 | 3.59 |
| SMILES-LSTM-HC | 6.88 | 5.60 | 6.88 | 6.88 | 6.88 | 6.88 | 6.88 | 6.88 | 6.88 | 6.88 |
| REINVENT | 3.55 | 3.85 | 3.85 | 3.73 | 3.39 | 3.39 | 3.39 | 3.39 | 4.04 | 3.39 |
| Pasithea | 3.35 | 3.48 | 3.24 | 3.38 | 3.24 | 3.24 | 3.24 | 3.24 | 3.24 | 3.24 |
| SMILES-VAE | 3.55 | 3.82 | 3.87 | 3.84 | 3.86 | 3.55 | 3.73 | 3.56 | 3.63 | 4.20 |
| graph-GA | 3.59 | 3.59 | 3.66 | 4.19 | 3.96 | 3.78 | 3.59 | 3.81 | 3.59 | 3.59 |
| MIMOSA | 4.15 | 4.15 | 4.15 | 4.15 | 4.15 | 4.15 | 4.15 | 4.15 | 4.15 | 4.15 |
| MolDQN | 0.49 | 1.00 | 1.07 | 1.07 | 0.56 | 1.00 | 1.00 | 0.29 | 1.00 | 1.45 |
| DST | 3.44 | 3.31 | 3.44 | 3.44 | 3.44 | 3.44 | 3.31 | 3.44 | 3.44 | 3.44 |
| JT-VAE | 3.93 | 3.91 | 3.91 | 4.37 | 4.70 | 3.75 | 4.10 | 3.75 | 3.75 | 4.36 |
| TargetDiff | – | -0.67 | – | 4.43 | 1.30 | 3.08 | 0.15 | -0.28 | – | – |
| Pocket2Mol | 5.49 | 3.72 | 5.23 | 6.31 | 4.22 | 3.30 | 4.27 | 4.04 | 3.27 | 5.04 |
| PocketFlow | 4.92 | 3.94 | 3.86 | 6.76 | 4.89 | 6.19 | 4.87 | 5.80 | 3.04 | 6.85 |
| ResGen | – | 3.29 | 2.18 | 3.75 | – | 2.90 | – | 2.90 | 2.79 | 2.90 |
| 3DSBDD | 2.41 | 3.51 | 0.72 | 3.30 | 1.23 | 2.01 | 6.93 | 3.79 | 1.14 | 2.67 |

| Model | 6GL8 | 1UWH | 7OTE | 1KKQ | 5WFD |
|---|---|---|---|---|---|
| SMILES-GA | 2.58 ± 0.44 | 2.55 ± 0.42 | 2.53 ± 0.44 | 2.43 ± 0.51 | 2.57 ± 0.43 |
| SMILES-LSTM-HC | 4.64 ± 1.03 | 4.52 ± 0.41 | 4.64 ± 1.03 | 4.64 ± 1.03 | 4.64 ± 1.03 |
| REINVENT | 3.14 ± 0.25 | 3.22 ± 0.27 | 3.13 ± 0.33 | 3.12 ± 0.29 | 3.02 ± 0.23 |
| Pasithea | 3.08 ± 0.12 | 3.04 ± 0.18 | 3.06 ± 0.09 | 3.06 ± 0.16 | 3.06 ± 0.09 |
| SMILES-VAE | 3.33 ± 0.09 | 3.36 ± 0.18 | 3.33 ± 0.26 | 3.34 ± 0.20 | 3.47 ± 0.19 |
| graph-GA | 2.78 ± 0.33 | 3.04 ± 0.32 | 3.00 ± 0.34 | 3.29 ± 0.51 | 3.20 ± 0.32 |
| MIMOSA | 3.34 ± 0.28 | 3.34 ± 0.28 | 3.34 ± 0.28 | 3.34 ± 0.28 | 3.34 ± 0.28 |
| MolDQN | 0.27 ± 0.20 | -0.13 ± 0.49 | 0.27 ± 0.49 | 0.50 ± 0.31 | 0.02 ± 0.32 |
| DST | 3.14 ± 0.14 | 3.09 ± 0.11 | 3.14 ± 0.14 | 3.14 ± 0.14 | 3.14 ± 0.14 |
| JT-VAE | 3.45 ± 0.28 | 3.54 ± 0.24 | 3.51 ± 0.20 | 3.71 ± 0.29 | 3.87 ± 0.46 |
| TargetDiff | – | -0.67 ± 0.00 | – | 3.79 ± 0.36 | -0.11 ± 0.87 |
| Pocket2Mol | 5.00 ± 0.21 | 3.33 ± 0.18 | 4.93 ± 0.18 | 6.00 ± 0.15 | 3.88 ± 0.21 |
| PocketFlow | 4.35 ± 0.27 | 2.29 ± 0.73 | 3.01 ± 0.58 | 6.03 ± 0.40 | 3.78 ± 0.49 |
| ResGen | – | 2.83 ± 0.18 | 1.78 ± 0.22 | 3.58 ± 0.08 | – |
| 3DSBDD | 1.98 ± 0.23 | 2.72 ± 0.47 | -0.59 ± 0.49 | 2.93 ± 0.24 | 0.33 ± 0.56 |

a): targets in CrossDocking

| Model | 7W7C | 8JJL | 7D42 | 7S1S | 6AZV |
|---|---|---|---|---|---|
| SMILES-GA | 2.55 ± 0.50 | 2.84 ± 0.28 | 2.59 ± 0.43 | 2.45 ± 0.49 | 2.54 ± 0.45 |
| SMILES-LSTM-HC | 4.64 ± 1.03 | 4.64 ± 1.03 | 4.64 ± 1.03 | 4.64 ± 1.03 | 4.64 ± 1.03 |
| REINVENT | 3.07 ± 0.22 | 3.13 ± 0.18 | 3.02 ± 0.23 | 3.38 ± 0.37 | 3.07 ± 0.21 |
| Pasithea | 3.06 ± 0.09 | 3.06 ± 0.09 | 3.06 ± 0.09 | 3.06 ± 0.09 | 3.06 ± 0.09 |
| SMILES-VAE | 3.31 ± 0.10 | 3.38 ± 0.15 | 3.35 ± 0.13 | 3.37 ± 0.17 | 3.32 ± 0.30 |
| graph-GA | 3.07 ± 0.39 | 2.92 ± 0.34 | 3.00 ± 0.40 | 3.03 ± 0.27 | 2.80 ± 0.29 |
| MIMOSA | 3.34 ± 0.28 | 3.34 ± 0.28 | 3.34 ± 0.28 | 3.34 ± 0.28 | 3.34 ± 0.28 |
| MolDQN | 0.15 ± 0.38 | -0.09 ± 0.54 | -0.23 ± 0.34 | 0.04 ± 0.36 | 0.06 ± 0.65 |
| DST | 3.14 ± 0.14 | 3.09 ± 0.11 | 3.14 ± 0.14 | 3.14 ± 0.14 | 3.14 ± 0.14 |
| JT-VAE | 2.94 ± 0.36 | 3.47 ± 0.34 | 3.38 ± 0.16 | 3.37 ± 0.18 | 3.58 ± 0.41 |
| TargetDiff | 1.01 ± 1.35 | -7.95 ± 20.45 | -0.28 ± 0.00 | – | – |
| Pocket2Mol | 2.78 ± 0.29 | 3.98 ± 0.13 | 3.54 ± 0.27 | 2.74 ± 0.23 | 4.38 ± 0.32 |
| PocketFlow | 5.50 ± 0.36 | 3.84 ± 0.51 | 5.48 ± 0.21 | 1.15 ± 0.79 | 6.40 ± 0.26 |
| ResGen | 2.59 ± 0.16 | – | 2.48 ± 0.19 | 2.36 ± 0.15 | 2.58 ± 0.20 |
| 3DSBDD | 0.91 ± 0.60 | 6.19 ± 0.35 | 2.94 ± 0.43 | -0.19 ± 0.64 | 1.71 ± 0.41 |

b): targets not in CrossDocking

Table 13: Top 10 LogP score for each target

| Model | 6GL8 | 1UWH | 7OTE | 1KKQ | 5WFD |
|---|---|---|---|---|---|
| SMILES-GA | 1.25 ± 0.62 | 1.36 ± 0.57 | 1.24 ± 0.63 | 0.93 ± 0.75 | 1.32 ± 0.63 |
| SMILES-LSTM-HC | 3.39 ± 0.60 | 3.38 ± 0.55 | 3.39 ± 0.60 | 3.39 ± 0.60 | 3.39 ± 0.60 |
| REINVENT | 2.21 ± 0.47 | 2.11 ± 0.58 | 2.12 ± 0.51 | 2.11 ± 0.53 | 2.08 ± 0.49 |
| Pasithea | 2.44 ± 0.35 | 2.40 ± 0.31 | 2.50 ± 0.30 | 2.42 ± 0.32 | 2.50 ± 0.30 |
| SMILES-VAE | 2.56 ± 0.40 | 2.55 ± 0.38 | 2.57 ± 0.34 | 2.45 ± 0.44 | 2.58 ± 0.42 |
| graph-GA | 1.61 ± 0.61 | 1.81 ± 0.58 | 1.64 ± 0.64 | 1.89 ± 0.65 | 1.99 ± 0.57 |
| MIMOSA | 2.55 ± 0.38 | 2.55 ± 0.38 | 2.55 ± 0.38 | 2.55 ± 0.38 | 2.55 ± 0.38 |
| MolDQN | -1.41 ± 0.72 | -1.67 ± 0.66 | -1.51 ± 0.75 | -1.17 ± 0.76 | -1.62 ± 0.71 |
| DST | 2.60 ± 0.28 | 2.57 ± 0.27 | 2.60 ± 0.28 | 2.60 ± 0.28 | 2.60 ± 0.28 |
| JT-VAE | 2.25 ± 0.54 | 2.54 ± 0.49 | 2.52 ± 0.47 | 2.47 ± 0.56 | 2.46 ± 0.63 |
| TargetDiff | – | -0.67 ± 0.00 | – | 2.14 ± 0.77 | -2.55 ± 1.65 |
| Pocket2Mol | 3.72 ± 0.68 | 2.59 ± 0.39 | 3.97 ± 0.44 | 4.78 ± 0.66 | 3.02 ± 0.41 |
| PocketFlow | 3.02 ± 0.65 | 0.57 ± 0.75 | 1.85 ± 0.54 | 4.40 ± 0.74 | 1.86 ± 0.88 |
| ResGen | – | 2.19 ± 0.32 | 0.77 ± 0.52 | 2.92 ± 0.32 | – |
| 3DSBDD | -0.19 ± 1.18 | 1.01 ± 0.75 | -2.16 ± 0.76 | 0.56 ± 1.15 | -2.56 ± 1.25 |

a): targets in CrossDocking

| Model | 7W7C | 8JJL | 7D42 | 7S1S | 6AZV |
|---|---|---|---|---|---|
| SMILES-GA | 0.91 ± 0.79 | 1.54 ± 0.60 | 1.03 ± 0.75 | 1.23 ± 0.65 | 1.28 ± 0.61 |
| SMILES-LSTM-HC | 3.39 ± 0.60 | 3.39 ± 0.60 | 3.39 ± 0.60 | 3.39 ± 0.60 | 3.39 ± 0.60 |
| REINVENT | 2.08 ± 0.53 | 2.16 ± 0.53 | 2.08 ± 0.50 | 2.21 ± 0.57 | 2.15 ± 0.50 |
| Pasithea | 2.50 ± 0.30 | 2.50 ± 0.30 | 2.50 ± 0.30 | 2.50 ± 0.30 | 2.50 ± 0.30 |
| SMILES-VAE | 2.54 ± 0.35 | 2.55 ± 0.40 | 2.57 ± 0.38 | 2.53 ± 0.39 | 2.47 ± 0.39 |
| graph-GA | 1.81 ± 0.62 | 1.58 ± 0.66 | 1.83 ± 0.55 | 1.69 ± 0.64 | 1.63 ± 0.62 |
| MIMOSA | 2.55 ± 0.38 | 2.55 ± 0.38 | 2.55 ± 0.38 | 2.55 ± 0.38 | 2.55 ± 0.38 |
| MolDQN | -1.14 ± 0.68 | -1.68 ± 0.70 | -1.83 ± 0.68 | -1.49 ± 0.69 | -1.63 ± 0.75 |
| DST | 2.60 ± 0.28 | 2.57 ± 0.27 | 2.60 ± 0.28 | 2.60 ± 0.28 | 2.60 ± 0.28 |
| JT-VAE | 1.13 ± 0.92 | 2.28 ± 0.59 | 2.50 ± 0.50 | 2.28 ± 0.53 | 2.35 ± 0.56 |
| TargetDiff | 1.01 ± 1.35 | -7.95 ± 20.45 | -0.28 ± 0.00 | – | – |
| Pocket2Mol | 1.62 ± 0.58 | 3.43 ± 0.27 | 2.05 ± 0.70 | 1.35 ± 0.67 | 3.24 ± 0.56 |
| PocketFlow | 4.30 ± 0.52 | 2.52 ± 0.64 | 4.51 ± 0.46 | -0.05 ± 0.64 | 5.24 ± 0.57 |
| ResGen | 1.63 ± 0.49 | – | 1.67 ± 0.39 | 1.62 ± 0.39 | 1.79 ± 0.36 |
| 3DSBDD | -11.36 ± 11.32 | 3.73 ± 1.18 | 0.04 ± 1.80 | -3.82 ± 1.76 | -0.32 ± 1.20 |

b): targets not in CrossDocking

Table 14: Top 100 LogP score for each target

Table 15: Top 1 QED score for each target. Targets in CrossDocking are marked in red, and targets not in CrossDocking are in blue.

| Model | 6GL8 | 1UWH | 7OTE | 1KKQ | 5WFD | 7W7C | 8JJL | 7D42 | 7S1S | 6AZV |
|---|---|---|---|---|---|---|---|---|---|---|
| SMILES-GA | 0.94 | 0.93 | 0.94 | 0.94 | 0.94 | 0.92 | 0.92 | 0.94 | 0.92 | 0.92 |
| SMILES-LSTM-HC | 0.94 | 0.94 | 0.94 | 0.94 | 0.94 | 0.94 | 0.94 | 0.94 | 0.94 | 0.94 |
| REINVENT | 0.95 | 0.95 | 0.95 | 0.95 | 0.95 | 0.95 | 0.95 | 0.95 | 0.95 | 0.95 |
| Pasithea | 0.95 | 0.95 | 0.94 | 0.94 | 0.94 | 0.94 | 0.94 | 0.94 | 0.94 | 0.94 |
| SMILES-VAE | 0.95 | 0.95 | 0.95 | 0.95 | 0.95 | 0.95 | 0.95 | 0.95 | 0.95 | 0.95 |
| graph-GA | 0.93 | 0.94 | 0.93 | 0.94 | 0.93 | 0.94 | 0.92 | 0.92 | 0.94 | 0.93 |
| MIMOSA | 0.95 | 0.95 | 0.95 | 0.95 | 0.95 | 0.95 | 0.95 | 0.95 | 0.95 | 0.95 |
| MolDQN | 0.65 | 0.75 | 0.58 | 0.63 | 0.59 | 0.55 | 0.66 | 0.53 | 0.72 | 0.55 |
| DST | 0.94 | 0.94 | 0.94 | 0.94 | 0.94 | 0.94 | 0.94 | 0.94 | 0.94 | 0.94 |
| JT-VAE | 0.94 | 0.94 | 0.94 | 0.94 | 0.94 | 0.94 | 0.94 | 0.94 | 0.94 | 0.94 |
| TargetDiff | – | 0.06 | – | 0.86 | 0.72 | 0.28 | 0.51 | 0.38 | – | – |
| Pocket2Mol | 0.95 | 0.94 | 0.94 | 0.94 | 0.93 | 0.94 | 0.93 | 0.89 | 0.94 | 0.94 |
| PocketFlow | 0.91 | 0.79 | 0.88 | 0.93 | 0.88 | 0.92 | 0.90 | 0.94 | 0.78 | 0.90 |
| ResGen | – | 0.78 | 0.65 | 0.93 | – | 0.70 | – | 0.89 | 0.93 | 0.89 |
| 3DSBDD | 0.92 | 0.92 | 0.88 | 0.86 | 0.85 | 0.91 | 0.87 | 0.93 | 0.91 | 0.92 |

| Model | 6GL8 | 1UWH | 7OTE | 1KKQ | 5WFD |
|---|---|---|---|---|---|
| SMILES-GA | $0.92 \pm 0.01$ | $0.91 \pm 0.01$ | $0.92 \pm 0.01$ | $0.93 \pm 0.01$ | $0.92 \pm 0.01$ |
| SMILES-LSTM-HC | $0.93 \pm 0.01$ | $0.93 \pm 0.01$ | $0.93 \pm 0.01$ | $0.93 \pm 0.01$ | $0.93 \pm 0.01$ |
| REINVENT | $0.94 \pm 0.01$ | $0.93 \pm 0.01$ | $0.93 \pm 0.01$ | $0.94 \pm 0.01$ | $0.94 \pm 0.01$ |
| Pasithea | $0.94 \pm 0.01$ | $0.94 \pm 0.01$ | $0.94 \pm 0.00$ | $0.93 \pm 0.00$ | $0.94 \pm 0.00$ |
| SMILES-VAE | $0.94 \pm 0.00$ | $0.94 \pm 0.00$ | $0.94 \pm 0.00$ | $0.94 \pm 0.01$ | $0.94 \pm 0.00$ |
| graph-GA | $0.92 \pm 0.01$ | $0.91 \pm 0.01$ | $0.92 \pm 0.01$ | $0.91 \pm 0.01$ | $0.91 \pm 0.01$ |
| MIMOSA | $0.94 \pm 0.00$ | $0.94 \pm 0.00$ | $0.94 \pm 0.00$ | $0.94 \pm 0.00$ | $0.94 \pm 0.00$ |
| MolDQN | $0.60 \pm 0.03$ | $0.67 \pm 0.03$ | $0.54 \pm 0.02$ | $0.60 \pm 0.02$ | $0.53 \pm 0.04$ |
| DST | $0.94 \pm 0.00$ | $0.94 \pm 0.00$ | $0.94 \pm 0.00$ | $0.94 \pm 0.00$ | $0.94 \pm 0.00$ |
| JT-VAE | $0.93 \pm 0.01$ | $0.93 \pm 0.01$ | $0.93 \pm 0.01$ | $0.92 \pm 0.01$ | $0.93 \pm 0.01$ |
| TargetDiff | – | $0.06 \pm 0.00$ | – | $0.82 \pm 0.02$ | $0.56 \pm 0.09$ |
| Pocket2Mol | $0.93 \pm 0.01$ | $0.92 \pm 0.01$ | $0.93 \pm 0.01$ | $0.91 \pm 0.02$ | $0.91 \pm 0.01$ |
| PocketFlow | $0.90 \pm 0.01$ | $0.74 \pm 0.03$ | $0.83 \pm 0.02$ | $0.90 \pm 0.02$ | $0.85 \pm 0.02$ |
| ResGen | – | $0.74 \pm 0.02$ | $0.63 \pm 0.01$ | $0.91 \pm 0.01$ | – |
| 3DSBDD | $0.90 \pm 0.01$ | $0.88 \pm 0.02$ | $0.82 \pm 0.03$ | $0.79 \pm 0.03$ | $0.79 \pm 0.03$ |

a): targets in CrossDocking

| Model | 7W7C | 8JJL | 7D42 | 7S1S | 6AZV |
|---|---|---|---|---|---|
| SMILES-GA | $0.91 \pm 0.01$ | $0.91 \pm 0.01$ | $0.92 \pm 0.01$ | $0.91 \pm 0.01$ | $0.91 \pm 0.01$ |
| SMILES-LSTM-HC | $0.93 \pm 0.01$ | $0.93 \pm 0.01$ | $0.93 \pm 0.01$ | $0.93 \pm 0.01$ | $0.93 \pm 0.01$ |
| REINVENT | $0.94 \pm 0.01$ | $0.93 \pm 0.01$ | $0.94 \pm 0.01$ | $0.93 \pm 0.01$ | $0.93 \pm 0.01$ |
| Pasithea | $0.94 \pm 0.00$ | $0.94 \pm 0.00$ | $0.94 \pm 0.00$ | $0.94 \pm 0.00$ | $0.94 \pm 0.00$ |
| SMILES-VAE | $0.94 \pm 0.00$ | $0.94 \pm 0.00$ | $0.94 \pm 0.00$ | $0.94 \pm 0.00$ | $0.94 \pm 0.00$ |
| graph-GA | $0.92 \pm 0.01$ | $0.90 \pm 0.01$ | $0.91 \pm 0.01$ | $0.91 \pm 0.01$ | $0.91 \pm 0.01$ |
| MIMOSA | $0.94 \pm 0.00$ | $0.94 \pm 0.00$ | $0.94 \pm 0.00$ | $0.94 \pm 0.00$ | $0.94 \pm 0.00$ |
| MolDQN | $0.52 \pm 0.01$ | $0.56 \pm 0.04$ | $0.50 \pm 0.02$ | $0.67 \pm 0.03$ | $0.52 \pm 0.01$ |
| DST | $0.94 \pm 0.00$ | $0.94 \pm 0.00$ | $0.94 \pm 0.00$ | $0.94 \pm 0.00$ | $0.94 \pm 0.00$ |
| JT-VAE | $0.91 \pm 0.01$ | $0.93 \pm 0.01$ | $0.92 \pm 0.01$ | $0.93 \pm 0.01$ | $0.93 \pm 0.01$ |
| TargetDiff | $0.17 \pm 0.09$ | $0.23 \pm 0.18$ | $0.38 \pm 0.00$ | – | – |
| Pocket2Mol | $0.93 \pm 0.00$ | $0.92 \pm 0.01$ | $0.87 \pm 0.01$ | $0.91 \pm 0.01$ | $0.93 \pm 0.01$ |
| PocketFlow | $0.89 \pm 0.02$ | $0.88 \pm 0.01$ | $0.87 \pm 0.03$ | $0.72 \pm 0.03$ | $0.88 \pm 0.01$ |
| ResGen | $0.67 \pm 0.01$ | – | $0.88 \pm 0.01$ | $0.91 \pm 0.01$ | $0.84 \pm 0.02$ |
| 3DSBDD | $0.78 \pm 0.06$ | $0.79 \pm 0.04$ | $0.90 \pm 0.02$ | $0.85 \pm 0.04$ | $0.90 \pm 0.02$ |

b): targets not in CrossDocking

Table 16: Top 10 QED score for each target

| Model | 6GL8 | 1UWH | 7OTE | 1KKQ | 5WFD |
|---|---|---|---|---|---|
| SMILES-GA | $0.87 \pm 0.03$ | $0.87 \pm 0.02$ | $0.87 \pm 0.03$ | $0.86 \pm 0.03$ | $0.86 \pm 0.03$ |
| SMILES-LSTM-HC | $0.88 \pm 0.03$ | $0.88 \pm 0.03$ | $0.88 \pm 0.03$ | $0.88 \pm 0.03$ | $0.88 \pm 0.03$ |
| REINVENT | $0.89 \pm 0.02$ | $0.89 \pm 0.02$ | $0.89 \pm 0.02$ | $0.89 \pm 0.02$ | $0.90 \pm 0.02$ |
| Pasithea | $0.90 \pm 0.02$ | $0.90 \pm 0.02$ | $0.90 \pm 0.02$ | $0.90 \pm 0.02$ | $0.90 \pm 0.02$ |
| SMILES-VAE | $0.91 \pm 0.01$ | $0.91 \pm 0.01$ | $0.91 \pm 0.02$ | $0.89 \pm 0.02$ | $0.91 \pm 0.02$ |
| graph-GA | $0.84 \pm 0.04$ | $0.84 \pm 0.04$ | $0.84 \pm 0.04$ | $0.84 \pm 0.04$ | $0.84 \pm 0.04$ |
| MIMOSA | $0.91 \pm 0.02$ | $0.91 \pm 0.02$ | $0.91 \pm 0.02$ | $0.91 \pm 0.02$ | $0.91 \pm 0.02$ |
| MolDQN | $0.45 \pm 0.06$ | $0.49 \pm 0.08$ | $0.44 \pm 0.05$ | $0.47 \pm 0.06$ | $0.44 \pm 0.05$ |
| DST | $0.91 \pm 0.02$ | $0.90 \pm 0.02$ | $0.91 \pm 0.02$ | $0.91 \pm 0.02$ | $0.91 \pm 0.02$ |
| JT-VAE | $0.87 \pm 0.03$ | $0.88 \pm 0.03$ | $0.88 \pm 0.02$ | $0.88 \pm 0.02$ | $0.87 \pm 0.03$ |
| TargetDiff | – | $0.06 \pm 0.00$ | – | $0.66 \pm 0.08$ | $0.24 \pm 0.17$ |
| Pocket2Mol | $0.86 \pm 0.04$ | $0.81 \pm 0.05$ | $0.84 \pm 0.05$ | $0.75 \pm 0.09$ | $0.83 \pm 0.05$ |
| PocketFlow | $0.78 \pm 0.06$ | $0.62 \pm 0.06$ | $0.74 \pm 0.05$ | $0.79 \pm 0.06$ | $0.73 \pm 0.06$ |
| ResGen | – | $0.68 \pm 0.03$ | $0.58 \pm 0.03$ | $0.87 \pm 0.02$ | – |
| 3DSBDD | $0.78 \pm 0.06$ | $0.76 \pm 0.07$ | $0.73 \pm 0.05$ | $0.64 \pm 0.09$ | $0.62 \pm 0.08$ |

a): targets in CrossDocking

| Model | 7W7C | 8JJL | 7D42 | 7S1S | 6AZV |
|---|---|---|---|---|---|
| SMILES-GA | $0.86 \pm 0.03$ | $0.86 \pm 0.03$ | $0.86 \pm 0.03$ | $0.86 \pm 0.03$ | $0.86 \pm 0.03$ |
| SMILES-LSTM-HC | $0.88 \pm 0.03$ | $0.88 \pm 0.03$ | $0.88 \pm 0.03$ | $0.88 \pm 0.03$ | $0.88 \pm 0.03$ |
| REINVENT | $0.89 \pm 0.02$ | $0.89 \pm 0.02$ | $0.89 \pm 0.02$ | $0.89 \pm 0.02$ | $0.89 \pm 0.02$ |
| Pasithea | $0.90 \pm 0.02$ | $0.90 \pm 0.02$ | $0.90 \pm 0.02$ | $0.90 \pm 0.02$ | $0.90 \pm 0.02$ |
| SMILES-VAE | $0.91 \pm 0.02$ | $0.91 \pm 0.02$ | $0.91 \pm 0.02$ | $0.91 \pm 0.02$ | $0.91 \pm 0.02$ |
| graph-GA | $0.84 \pm 0.04$ | $0.83 \pm 0.04$ | $0.84 \pm 0.04$ | $0.84 \pm 0.04$ | $0.83 \pm 0.04$ |
| MIMOSA | $0.91 \pm 0.02$ | $0.91 \pm 0.02$ | $0.91 \pm 0.02$ | $0.91 \pm 0.02$ | $0.91 \pm 0.02$ |
| MolDQN | $0.44 \pm 0.04$ | $0.45 \pm 0.05$ | $0.43 \pm 0.03$ | $0.49 \pm 0.08$ | $0.43 \pm 0.04$ |
| DST | $0.91 \pm 0.02$ | $0.90 \pm 0.02$ | $0.91 \pm 0.02$ | $0.91 \pm 0.02$ | $0.91 \pm 0.02$ |
| JT-VAE | $0.82 \pm 0.05$ | $0.88 \pm 0.03$ | $0.87 \pm 0.02$ | $0.88 \pm 0.03$ | $0.88 \pm 0.03$ |
| TargetDiff | $0.17 \pm 0.09$ | $0.23 \pm 0.18$ | $0.38 \pm 0.00$ | – | – |
| Pocket2Mol | $0.89 \pm 0.03$ | $0.87 \pm 0.03$ | $0.81 \pm 0.03$ | $0.83 \pm 0.04$ | $0.84 \pm 0.05$ |
| PocketFlow | $0.80 \pm 0.05$ | $0.79 \pm 0.05$ | $0.75 \pm 0.06$ | $0.60 \pm 0.06$ | $0.78 \pm 0.05$ |
| ResGen | $0.62 \pm 0.03$ | – | $0.80 \pm 0.04$ | $0.87 \pm 0.02$ | $0.78 \pm 0.04$ |
| 3DSBDD | $0.49 \pm 0.16$ | $0.62 \pm 0.07$ | $0.75 \pm 0.08$ | $0.52 \pm 0.18$ | $0.73 \pm 0.09$ |

b): targets not in CrossDocking

Table 17: Top 100 QED score for each target

Table 18: Top 1 SA score for each target. Targets in CrossDocking are marked in red, and targets not in CrossDocking are in blue.

| Model | 6GL8 | 1UWH | 7OTE | 1KKQ | 5WFD | 7W7C | 8JJL | 7D42 | 7S1S | 6AZV |
|---|---|---|---|---|---|---|---|---|---|---|
| SMILES-GA | 1.65 | 1.65 | 1.65 | 1.65 | 1.65 | 1.65 | 1.65 | 1.65 | 1.65 | 1.65 |
| SMILES-LSTM-HC | 1.51 | 1.50 | 1.51 | 1.51 | 1.51 | 1.51 | 1.51 | 1.51 | 1.51 | 1.51 |
| REINVENT | 1.58 | 1.58 | 1.58 | 1.58 | 1.58 | 1.58 | 1.27 | 1.58 | 1.58 | 1.58 |
| Pasithea | 1.63 | 1.52 | 1.50 | 1.54 | 1.50 | 1.50 | 1.50 | 1.50 | 1.50 | 1.50 |
| SMILES-VAE | 1.44 | 1.50 | 1.50 | 1.50 | 1.50 | 1.50 | 1.44 | 1.50 | 1.50 | 1.50 |
| graph-GA | 1.00 | 1.00 | 1.00 | 1.00 | 1.00 | 1.00 | 1.00 | 1.00 | 1.00 | 1.00 |
| MIMOSA | 1.50 | 1.50 | 1.50 | 1.50 | 1.50 | 1.50 | 1.50 | 1.50 | 1.50 | 1.50 |
| MolDQN | 1.51 | 1.75 | 1.65 | 1.51 | 1.85 | 1.62 | 1.62 | 1.85 | 1.51 | 1.61 |
| DST | 1.50 | 1.50 | 1.50 | 1.50 | 1.50 | 1.50 | 1.50 | 1.50 | 1.50 | 1.50 |
| JT-VAE | 1.50 | 1.38 | 1.50 | 1.38 | 1.25 | 1.50 | 1.50 | 1.11 | 1.50 | 1.41 |
| TargetDiff | − | 4.05 | − | 1.76 | 2.96 | 3.86 | 3.68 | 3.41 | − | − |
| Pocket2Mol | 1.00 | 1.03 | 1.00 | 1.03 | 1.00 | 1.09 | 1.18 | 1.63 | 1.32 | 1.00 |
| PocketFlow | 1.00 | 1.00 | 1.00 | 1.00 | 1.00 | 1.00 | 1.00 | 1.00 | 1.61 | 1.00 |
| ResGen | − | 1.00 | 1.18 | 1.00 | − | 1.00 | − | 1.00 | 1.16 | 1.00 |
| 3DSBDD | 1.00 | 1.20 | 2.03 | 1.84 | 1.00 | 3.07 | 1.21 | 1.55 | 1.88 | 1.00 |

| Model | 6GL8 | 1UWH | 7OTE | 1KKQ | 5WFD |
|---|---|---|---|---|---|
| SMILES-GA | 1.84 ± 0.10 | 1.87 ± 0.12 | 1.79 ± 0.08 | 1.87 ± 0.12 | 1.85 ± 0.11 |
| SMILES-LSTM-HC | 1.67 ± 0.07 | 1.66 ± 0.08 | 1.67 ± 0.07 | 1.67 ± 0.07 | 1.67 ± 0.07 |
| REINVENT | 1.72 ± 0.07 | 1.73 ± 0.07 | 1.74 ± 0.08 | 1.74 ± 0.08 | 1.74 ± 0.08 |
| Pasithea | 1.73 ± 0.05 | 1.70 ± 0.11 | 1.66 ± 0.09 | 1.67 ± 0.07 | 1.66 ± 0.09 |
| SMILES-VAE | 1.64 ± 0.10 | 1.67 ± 0.07 | 1.67 ± 0.08 | 1.64 ± 0.10 | 1.64 ± 0.08 |
| graph-GA | 1.24 ± 0.25 | 1.07 ± 0.09 | 1.35 ± 0.24 | 1.12 ± 0.13 | 1.08 ± 0.05 |
| MIMOSA | 1.70 ± 0.08 | 1.70 ± 0.08 | 1.70 ± 0.08 | 1.70 ± 0.08 | 1.70 ± 0.08 |
| MolDQN | 1.84 ± 0.14 | 2.09 ± 0.19 | 1.90 ± 0.18 | 1.83 ± 0.14 | 2.02 ± 0.15 |
| DST | 1.66 ± 0.09 | 1.66 ± 0.09 | 1.66 ± 0.09 | 1.66 ± 0.09 | 1.66 ± 0.09 |
| JT-VAE | 1.81 ± 0.12 | 1.71 ± 0.15 | 1.67 ± 0.09 | 1.65 ± 0.13 | 1.64 ± 0.17 |
| TargetDiff | – | 4.05 ± 0.00 | – | 2.41 ± 0.27 | 3.54 ± 0.28 |
| Pocket2Mol | 1.20 ± 0.13 | 1.50 ± 0.22 | 1.10 ± 0.08 | 1.45 ± 0.20 | 1.05 ± 0.06 |
| PocketFlow | 1.00 ± 0.00 | 1.37 ± 0.26 | 1.00 ± 0.00 | 1.07 ± 0.06 | 1.11 ± 0.10 |
| ResGen | – | 1.00 ± 0.00 | 1.40 ± 0.12 | 1.05 ± 0.06 | – |
| 3DSBDD | 1.44 ± 0.25 | 1.47 ± 0.16 | 2.92 ± 0.41 | 2.90 ± 0.39 | 2.02 ± 0.59 |

a): targets in CrossDocking

| Model | 7W7C | 8JJL | 7D42 | 7S1S | 6AZV |
|---|---|---|---|---|---|
| SMILES-GA | 1.89 ± 0.14 | 1.85 ± 0.11 | 1.87 ± 0.11 | 1.86 ± 0.11 | 1.85 ± 0.09 |
| SMILES-LSTM-HC | 1.67 ± 0.07 | 1.67 ± 0.07 | 1.67 ± 0.07 | 1.67 ± 0.07 | 1.67 ± 0.07 |
| REINVENT | 1.74 ± 0.08 | 1.67 ± 0.15 | 1.74 ± 0.08 | 1.73 ± 0.07 | 1.73 ± 0.08 |
| Pasithea | 1.66 ± 0.09 | 1.66 ± 0.09 | 1.66 ± 0.09 | 1.66 ± 0.09 | 1.66 ± 0.09 |
| SMILES-VAE | 1.71 ± 0.09 | 1.64 ± 0.10 | 1.65 ± 0.07 | 1.67 ± 0.09 | 1.70 ± 0.08 |
| graph-GA | 1.12 ± 0.14 | 1.30 ± 0.21 | 1.14 ± 0.18 | 1.15 ± 0.16 | 1.31 ± 0.25 |
| MIMOSA | 1.70 ± 0.08 | 1.70 ± 0.08 | 1.70 ± 0.08 | 1.70 ± 0.08 | 1.70 ± 0.08 |
| MolDQN | 1.85 ± 0.16 | 2.01 ± 0.27 | 2.08 ± 0.20 | 1.92 ± 0.18 | 1.96 ± 0.20 |
| DST | 1.66 ± 0.09 | 1.66 ± 0.09 | 1.66 ± 0.09 | 1.66 ± 0.09 | 1.66 ± 0.09 |
| JT-VAE | 1.88 ± 0.13 | 1.83 ± 0.14 | 1.70 ± 0.24 | 1.78 ± 0.16 | 1.60 ± 0.10 |
| TargetDiff | 4.59 ± 0.67 | 5.23 ± 0.86 | 3.41 ± 0.00 | – | – |
| Pocket2Mol | 1.29 ± 0.13 | 1.36 ± 0.12 | 2.10 ± 0.22 | 1.78 ± 0.29 | 1.54 ± 0.22 |
| PocketFlow | 1.10 ± 0.08 | 1.16 ± 0.09 | 1.00 ± 0.00 | 1.77 ± 0.08 | 1.00 ± 0.00 |
| ResGen | 1.15 ± 0.11 | – | 1.14 ± 0.10 | 1.43 ± 0.11 | 1.05 ± 0.06 |
| 3DSBDD | 4.40 ± 0.57 | 2.76 ± 0.76 | 1.98 ± 0.27 | 2.65 ± 0.48 | 1.80 ± 0.41 |

b): targets not in CrossDocking

Table 19: Top 10 SA score for each target

| Model | 6GL8 | 1UWH | 7OTE | 1KKQ | 5WFD |
|---|---|---|---|---|---|
| SMILES-GA | $2.55 \pm 0.35$ | $2.50 \pm 0.33$ | $2.53 \pm 0.37$ | $2.77 \pm 0.49$ | $2.52 \pm 0.37$ |
| SMILES-LSTM-HC | $1.93 \pm 0.13$ | $1.95 \pm 0.14$ | $1.93 \pm 0.13$ | $1.93 \pm 0.13$ | $1.93 \pm 0.13$ |
| REINVENT | $2.14 \pm 0.20$ | $2.14 \pm 0.19$ | $2.14 \pm 0.19$ | $2.15 \pm 0.19$ | $2.14 \pm 0.18$ |
| Pasithea | $2.01 \pm 0.14$ | $2.00 \pm 0.14$ | $1.97 \pm 0.15$ | $1.97 \pm 0.14$ | $1.96 \pm 0.15$ |
| SMILES-VAE | $1.96 \pm 0.14$ | $1.94 \pm 0.13$ | $1.96 \pm 0.14$ | $2.02 \pm 0.17$ | $1.96 \pm 0.15$ |
| graph-GA | $2.10 \pm 0.36$ | $1.94 \pm 0.39$ | $2.15 \pm 0.37$ | $1.94 \pm 0.35$ | $1.96 \pm 0.43$ |
| MIMOSA | $1.95 \pm 0.12$ | $1.95 \pm 0.12$ | $1.95 \pm 0.12$ | $1.95 \pm 0.12$ | $1.95 \pm 0.12$ |
| MolDQN | $2.93 \pm 0.57$ | $3.21 \pm 0.51$ | $2.98 \pm 0.50$ | $2.83 \pm 0.53$ | $3.15 \pm 0.53$ |
| DST | $1.95 \pm 0.14$ | $1.96 \pm 0.14$ | $1.95 \pm 0.14$ | $1.95 \pm 0.14$ | $1.95 \pm 0.14$ |
| JT-VAE | $2.29 \pm 0.25$ | $2.19 \pm 0.22$ | $2.21 \pm 0.24$ | $2.21 \pm 0.25$ | $2.23 \pm 0.27$ |
| TargetDiff | $-$ | $4.05 \pm 0.00$ | $-$ | $3.42 \pm 0.48$ | $4.50 \pm 0.63$ |
| Pocket2Mol | $1.69 \pm 0.25$ | $2.07 \pm 0.29$ | $1.92 \pm 0.37$ | $2.05 \pm 0.29$ | $1.43 \pm 0.18$ |
| PocketFlow | $1.36 \pm 0.22$ | $1.98 \pm 0.30$ | $1.30 \pm 0.23$ | $1.60 \pm 0.29$ | $1.86 \pm 0.34$ |
| ResGen | $-$ | $1.29 \pm 0.17$ | $1.87 \pm 0.25$ | $1.49 \pm 0.21$ | $-$ |
| 3DSBDD | $3.00 \pm 0.74$ | $2.89 \pm 0.73$ | $4.39 \pm 0.65$ | $4.20 \pm 0.61$ | $4.40 \pm 1.03$ |

a): targets in CrossDocking

| Model | 7W7C | 8JJL | 7D42 | 7S1S | 6AZV |
|---|---|---|---|---|---|
| SMILES-GA | $2.77 \pm 0.48$ | $2.48 \pm 0.32$ | $2.66 \pm 0.43$ | $2.53 \pm 0.35$ | $2.55 \pm 0.38$ |
| SMILES-LSTM-HC | $1.93 \pm 0.13$ | $1.93 \pm 0.13$ | $1.93 \pm 0.13$ | $1.93 \pm 0.13$ | $1.93 \pm 0.13$ |
| REINVENT | $2.16 \pm 0.20$ | $2.14 \pm 0.23$ | $2.16 \pm 0.19$ | $2.11 \pm 0.18$ | $2.13 \pm 0.19$ |
| Pasithea | $1.97 \pm 0.15$ | $1.97 \pm 0.15$ | $1.97 \pm 0.15$ | $1.97 \pm 0.15$ | $1.97 \pm 0.15$ |
| SMILES-VAE | $1.99 \pm 0.13$ | $1.95 \pm 0.14$ | $1.95 \pm 0.14$ | $1.97 \pm 0.13$ | $1.96 \pm 0.13$ |
| graph-GA | $1.97 \pm 0.37$ | $2.15 \pm 0.38$ | $2.00 \pm 0.37$ | $2.01 \pm 0.38$ | $2.13 \pm 0.35$ |
| MIMOSA | $1.95 \pm 0.12$ | $1.95 \pm 0.12$ | $1.95 \pm 0.12$ | $1.95 \pm 0.12$ | $1.95 \pm 0.12$ |
| MolDQN | $3.05 \pm 0.53$ | $3.24 \pm 0.55$ | $3.28 \pm 0.53$ | $3.05 \pm 0.53$ | $3.14 \pm 0.51$ |
| DST | $1.95 \pm 0.14$ | $1.96 \pm 0.14$ | $1.95 \pm 0.14$ | $1.95 \pm 0.14$ | $1.95 \pm 0.14$ |
| JT-VAE | $2.64 \pm 0.43$ | $2.30 \pm 0.22$ | $2.23 \pm 0.25$ | $2.30 \pm 0.26$ | $2.18 \pm 0.26$ |
| TargetDiff | $4.59 \pm 0.67$ | $5.23 \pm 0.86$ | $3.41 \pm 0.00$ | $-$ | $-$ |
| Pocket2Mol | $2.05 \pm 0.35$ | $2.19 \pm 0.39$ | $2.72 \pm 0.28$ | $2.57 \pm 0.39$ | $2.14 \pm 0.29$ |
| PocketFlow | $1.71 \pm 0.30$ | $1.62 \pm 0.26$ | $1.17 \pm 0.21$ | $2.02 \pm 0.10$ | $1.06 \pm 0.09$ |
| ResGen | $1.76 \pm 0.35$ | $-$ | $1.63 \pm 0.24$ | $1.86 \pm 0.21$ | $1.47 \pm 0.21$ |
| 3DSBDD | $6.10 \pm 0.96$ | $4.89 \pm 0.88$ | $3.73 \pm 0.90$ | $4.87 \pm 1.05$ | $3.57 \pm 0.83$ |

b): targets not in CrossDocking

Table 20: Top 100 SA score for each target

Table 21: Top 1 SC score for each target. Targets in CrossDocking are marked in red, and targets not in CrossDocking are in blue.

| Model | 6GL8 | 1UWH | 7OTE | 1KKQ | 5WFD | 7W7C | 8JJL | 7D42 | 7S1S | 6AZV |
|---|---|---|---|---|---|---|---|---|---|---|
| Pocket2Mol | 1.27 | 1.57 | 1.00 | 1.18 | 1.16 | 1.26 | 1.38 | 1.23 | 1.26 | 1.30 |
| PocketFlow | 1.02 | 1.01 | 1.02 | 1.00 | 1.02 | 1.02 | 1.02 | 1.02 | 1.05 | 1.02 |
| ResGen | – | 1.00 | 1.00 | 1.19 | – | 1.15 | – | 1.13 | 1.13 | 1.12 |
| 3DSBDD | 1.19 | 1.01 | 2.31 | 1.30 | 1.02 | 2.84 | 1.24 | 1.51 | 1.20 | 1.01 |
| dst | 1.56 | 1.56 | 1.56 | 1.56 | 1.56 | 1.56 | 1.56 | 1.56 | 1.56 | 1.56 |
| graph_ga | 1.00 | 1.00 | 1.23 | 1.00 | 1.00 | 1.00 | 1.02 | 1.00 | 1.00 | 1.00 |
| jt_vae | 1.00 | 1.00 | 1.08 | 1.00 | 1.00 | 1.74 | 1.00 | 1.08 | 1.01 | 1.00 |
| mimosa | 1.36 | 1.36 | 1.36 | 1.36 | 1.36 | 1.36 | 1.36 | 1.36 | 1.36 | 1.36 |
| moldqn | 1.00 | 1.00 | 1.00 | 1.00 | 1.00 | 1.00 | 1.00 | 1.00 | 1.00 | 1.00 |
| pasithea | 1.00 | 1.71 | 1.09 | 1.00 | 1.00 | 1.56 | 1.09 | 1.56 | 1.00 | 1.00 |
| reinvent | 1.22 | 1.00 | 1.26 | 1.00 | 1.26 | 1.00 | 1.15 | 1.19 | 1.01 | 1.00 |
| smiles_ga | 1.74 | 1.98 | 1.12 | 1.28 | 1.55 | 1.28 | 1.28 | 1.28 | 1.81 | 1.28 |
| smiles_lstm_hc | 1.20 | 1.08 | 1.20 | 1.20 | 1.20 | 1.20 | 1.20 | 1.20 | 1.20 | 1.20 |
| smiles_vae | 1.00 | 1.00 | 1.34 | 1.96 | 1.00 | 1.34 | 1.00 | 1.26 | 1.22 | 1.35 |
| targetdiff | – | 3.30 | – | 2.63 | 2.62 | 3.45 | 2.11 | 2.86 | – | – |

| Model | 6GL8 | 1UWH | 7OTE | 1KKQ | 5WFD |
|---|---|---|---|---|---|
| Pocket2Mol | 1.48 ± 0.10 | 2.11 ± 0.27 | 1.40 ± 0.21 | 1.90 ± 0.47 | 1.23 ± 0.03 |
| PocketFlow | 1.02 ± 0.00 | 1.02 ± 0.01 | 1.02 ± 0.00 | 1.14 ± 0.08 | 1.09 ± 0.06 |
| ResGen | – | 1.09 ± 0.07 | 1.09 ± 0.05 | 1.27 ± 0.05 | – |
| SBDD3D | 1.58 ± 0.32 | 1.69 ± 0.37 | 2.89 ± 0.40 | 2.48 ± 0.45 | 1.59 ± 0.27 |
| dst | 1.82 ± 0.16 | 1.86 ± 0.16 | 1.82 ± 0.16 | 1.82 ± 0.16 | 1.82 ± 0.16 |
| graph_ga | 1.01 ± 0.02 | 1.01 ± 0.01 | 1.33 ± 0.07 | 1.05 ± 0.05 | 1.00 ± 0.01 |
| jt_vae | 1.10 ± 0.07 | 1.14 ± 0.11 | 1.19 ± 0.08 | 1.11 ± 0.07 | 1.14 ± 0.08 |
| mimosa | 1.59 ± 0.15 | 1.59 ± 0.15 | 1.59 ± 0.15 | 1.59 ± 0.15 | 1.59 ± 0.15 |
| moldqn | 1.02 ± 0.03 | 1.01 ± 0.01 | 1.01 ± 0.02 | 1.00 ± 0.00 | 1.00 ± 0.01 |
| pasithea | 1.05 ± 0.07 | 1.96 ± 0.13 | 1.23 ± 0.10 | 1.15 ± 0.11 | 1.09 ± 0.06 |
| reinvent | 1.37 ± 0.13 | 1.45 ± 0.17 | 1.45 ± 0.08 | 1.35 ± 0.15 | 1.37 ± 0.09 |
| smiles_ga | 1.98 ± 0.09 | 2.02 ± 0.04 | 1.67 ± 0.37 | 1.95 ± 0.26 | 1.95 ± 0.17 |
| smiles_lstm_hc | 1.51 ± 0.16 | 1.57 ± 0.20 | 1.51 ± 0.16 | 1.51 ± 0.16 | 1.51 ± 0.16 |
| smiles_vae | 1.50 ± 0.29 | 1.53 ± 0.21 | 1.59 ± 0.11 | 2.08 ± 0.07 | 1.46 ± 0.18 |
| targetdiff | – | 3.30 ± 0.00 | – | 2.93 ± 0.16 | 3.04 ± 0.22 |

a): targets in CrossDocking

| Model | 7W7C | 8JJL | 7D42 | 7S1S | 6AZV |
|---|---|---|---|---|---|
| Pocket2Mol | 1.36 ± 0.12 | 1.71 ± 0.19 | 2.05 ± 0.58 | 2.41 ± 0.52 | 1.71 ± 0.20 |
| PocketFlow | 1.19 ± 0.10 | 1.11 ± 0.05 | 1.16 ± 0.07 | 1.05 ± 0.00 | 1.09 ± 0.06 |
| ResGen | 1.31 ± 0.08 | – | 1.22 ± 0.05 | 1.39 ± 0.11 | 1.19 ± 0.04 |
| SBDD3D | 3.11 ± 0.14 | 2.24 ± 0.60 | 1.67 ± 0.21 | 2.39 ± 0.69 | 1.41 ± 0.23 |
| dst | 1.82 ± 0.16 | 1.86 ± 0.16 | 1.82 ± 0.16 | 1.82 ± 0.16 | 1.82 ± 0.16 |
| graph_ga | 1.04 ± 0.05 | 1.27 ± 0.11 | 1.01 ± 0.02 | 1.02 ± 0.02 | 1.13 ± 0.08 |
| jt_vae | 2.26 ± 0.24 | 1.14 ± 0.11 | 1.15 ± 0.06 | 1.17 ± 0.10 | 1.07 ± 0.05 |
| mimosa | 1.59 ± 0.15 | 1.59 ± 0.15 | 1.58 ± 0.13 | 1.59 ± 0.15 | 1.59 ± 0.15 |
| moldqn | 1.00 ± 0.01 | 1.02 ± 0.03 | 1.02 ± 0.03 | 1.00 ± 0.01 | 1.01 ± 0.02 |
| pasithea | 1.87 ± 0.17 | 1.23 ± 0.10 | 1.87 ± 0.17 | 1.15 ± 0.09 | 1.16 ± 0.09 |
| reinvent | 1.30 ± 0.17 | 1.49 ± 0.16 | 1.41 ± 0.09 | 1.39 ± 0.18 | 1.36 ± 0.16 |
| smiles_ga | 1.62 ± 0.29 | 1.87 ± 0.26 | 1.61 ± 0.32 | 2.04 ± 0.09 | 1.89 ± 0.26 |
| smiles_lstm_hc | 1.51 ± 0.16 | 1.51 ± 0.16 | 1.51 ± 0.16 | 1.51 ± 0.16 | 1.51 ± 0.16 |
| smiles_vae | 1.56 ± 0.13 | 1.48 ± 0.24 | 1.57 ± 0.15 | 1.60 ± 0.16 | 1.65 ± 0.14 |
| targetdiff | 4.09 ± 0.45 | 3.65 ± 0.68 | 2.86 ± 0.00 | – | – |

b): targets not in CrossDocking

Table 22: Top 10 SC score for each target

| Model | 6GL8 | 1UWH | 7OTE | 1KKQ | 5WFD |
|---|---|---|---|---|---|
| Pocket2Mol | $2.17 \pm 0.40$ | $3.13 \pm 0.50$ | $2.76 \pm 0.69$ | $3.05 \pm 0.50$ | $1.61 \pm 0.21$ |
| PocketFlow | $1.10 \pm 0.09$ | $1.25 \pm 0.10$ | $1.15 \pm 0.08$ | $1.30 \pm 0.10$ | $1.27 \pm 0.08$ |
| ResGen | $-$ | $1.24 \pm 0.07$ | $1.35 \pm 0.13$ | $1.68 \pm 0.22$ | $-$ |
| SBDD3D | $2.60 \pm 0.46$ | $3.44 \pm 0.80$ | $3.69 \pm 0.38$ | $3.44 \pm 0.43$ | $3.08 \pm 0.74$ |
| dst | $2.39 \pm 0.26$ | $2.43 \pm 0.26$ | $2.39 \pm 0.26$ | $2.39 \pm 0.26$ | $2.39 \pm 0.26$ |
| graph_ga | $1.28 \pm 0.13$ | $1.25 \pm 0.13$ | $2.07 \pm 0.39$ | $1.32 \pm 0.13$ | $1.26 \pm 0.12$ |
| jt_vae | $1.73 \pm 0.37$ | $1.72 \pm 0.33$ | $1.77 \pm 0.32$ | $1.74 \pm 0.35$ | $1.76 \pm 0.34$ |
| mimosa | $2.12 \pm 0.23$ | $2.12 \pm 0.23$ | $2.12 \pm 0.23$ | $2.12 \pm 0.23$ | $2.12 \pm 0.23$ |
| moldqn | $1.24 \pm 0.10$ | $1.22 \pm 0.11$ | $1.22 \pm 0.11$ | $1.18 \pm 0.09$ | $1.22 \pm 0.11$ |
| pasithea | $1.67 \pm 0.33$ | $2.46 \pm 0.25$ | $1.85 \pm 0.30$ | $1.64 \pm 0.27$ | $1.70 \pm 0.31$ |
| reinvent | $1.91 \pm 0.24$ | $1.92 \pm 0.23$ | $1.91 \pm 0.22$ | $1.84 \pm 0.23$ | $1.87 \pm 0.24$ |
| smiles_ga | $2.36 \pm 0.21$ | $2.44 \pm 0.22$ | $2.27 \pm 0.29$ | $2.46 \pm 0.25$ | $2.38 \pm 0.23$ |
| smiles_lstm_hc | $2.03 \pm 0.25$ | $2.04 \pm 0.21$ | $2.03 \pm 0.25$ | $2.03 \pm 0.25$ | $2.03 \pm 0.25$ |
| smiles_vae | $2.10 \pm 0.25$ | $2.06 \pm 0.23$ | $2.06 \pm 0.22$ | $2.57 \pm 0.23$ | $2.01 \pm 0.24$ |
| targetdiff | $-$ | $3.30 \pm 0.00$ | $-$ | $3.47 \pm 0.25$ | $3.89 \pm 0.55$ |

a): targets in CrossDocking

| Model | 7W7C | 8JJL | 7D42 | 7S1S | 6AZV |
|---|---|---|---|---|---|
| Pocket2Mol | $2.11 \pm 0.43$ | $2.89 \pm 0.55$ | $2.92 \pm 0.37$ | $3.45 \pm 0.45$ | $2.87 \pm 0.53$ |
| PocketFlow | $1.30 \pm 0.07$ | $1.27 \pm 0.09$ | $1.27 \pm 0.14$ | $1.14 \pm 0.03$ | $1.20 \pm 0.05$ |
| ResGen | $1.77 \pm 0.27$ | $-$ | $1.49 \pm 0.15$ | $1.93 \pm 0.24$ | $1.40 \pm 0.13$ |
| SBDD3D | $4.17 \pm 0.53$ | $3.32 \pm 0.44$ | $2.70 \pm 0.50$ | $3.80 \pm 0.63$ | $3.03 \pm 0.75$ |
| dst | $2.39 \pm 0.26$ | $2.43 \pm 0.26$ | $2.39 \pm 0.26$ | $2.39 \pm 0.26$ | $2.39 \pm 0.26$ |
| graph_ga | $1.29 \pm 0.12$ | $1.88 \pm 0.32$ | $1.30 \pm 0.14$ | $1.30 \pm 0.13$ | $1.33 \pm 0.11$ |
| jt_vae | $3.13 \pm 0.40$ | $1.73 \pm 0.34$ | $1.72 \pm 0.33$ | $1.72 \pm 0.32$ | $1.68 \pm 0.34$ |
| mimosa | $2.12 \pm 0.23$ | $2.12 \pm 0.23$ | $2.11 \pm 0.23$ | $2.12 \pm 0.23$ | $2.12 \pm 0.23$ |
| moldqn | $1.20 \pm 0.10$ | $1.26 \pm 0.12$ | $1.26 \pm 0.12$ | $1.19 \pm 0.10$ | $1.22 \pm 0.11$ |
| pasithea | $2.44 \pm 0.26$ | $1.85 \pm 0.30$ | $2.44 \pm 0.26$ | $1.70 \pm 0.28$ | $1.70 \pm 0.28$ |
| reinvent | $1.84 \pm 0.26$ | $1.97 \pm 0.23$ | $1.86 \pm 0.22$ | $1.89 \pm 0.23$ | $1.86 \pm 0.24$ |
| smiles_ga | $2.31 \pm 0.30$ | $2.36 \pm 0.26$ | $2.21 \pm 0.27$ | $2.45 \pm 0.20$ | $2.37 \pm 0.24$ |
| smiles_lstm_hc | $2.03 \pm 0.25$ | $2.03 \pm 0.25$ | $2.03 \pm 0.25$ | $2.03 \pm 0.25$ | $2.03 \pm 0.25$ |
| smiles_vae | $2.06 \pm 0.23$ | $2.08 \pm 0.26$ | $2.09 \pm 0.23$ | $2.08 \pm 0.21$ | $2.10 \pm 0.21$ |
| targetdiff | $4.09 \pm 0.45$ | $3.65 \pm 0.68$ | $2.86 \pm 0.00$ | $-$ | $-$ |

b): targets not in CrossDocking

Table 23: Top 100 SC score for each target

Table 24: Top 1 RA score for each target. Targets in CrossDocking are marked in red, and targets not in CrossDocking are in blue.

| Model | 6GL8 | 1UWH | 7OTE | 1KKQ | 5WFD | 7W7C | 8JJL | 7D42 | 7S1S | 6AZV |
|---|---|---|---|---|---|---|---|---|---|---|
| Pocket2Mol | 1.00 | 1.00 | 1.00 | 1.00 | 1.00 | 1.00 | 1.00 | 1.00 | 1.00 | 1.00 |
| PocketFlow | 1.00 | 1.00 | 1.00 | 1.00 | 1.00 | 1.00 | 1.00 | 1.00 | 1.00 | 1.00 |
| ResGen | – | 1.00 | 1.00 | 1.00 | – | 1.00 | – | 1.00 | 1.00 | 1.00 |
| 3DSBDD | 1.00 | 1.00 | 0.99 | 0.98 | 0.99 | 0.98 | 0.98 | 0.99 | 0.99 | 1.00 |
| dst | 1.00 | 1.00 | 1.00 | 1.00 | 1.00 | 1.00 | 1.00 | 1.00 | 1.00 | 1.00 |
| graph_ga | 1.00 | 1.00 | 0.99 | 1.00 | 1.00 | 1.00 | 1.00 | 1.00 | 1.00 | 1.00 |
| jt_vae | 1.00 | 1.00 | 1.00 | 1.00 | 1.00 | 1.00 | 1.00 | 1.00 | 1.00 | 1.00 |
| mimosa | 1.00 | 1.00 | 1.00 | 1.00 | 1.00 | 1.00 | 1.00 | 1.00 | 1.00 | 1.00 |
| moldqn | 0.99 | 1.00 | 1.00 | 0.99 | 0.99 | 0.99 | 0.99 | 0.99 | 0.99 | 0.99 |
| pasithea | 1.00 | 1.00 | 1.00 | 1.00 | 1.00 | 1.00 | 1.00 | 1.00 | 1.00 | 1.00 |
| reinvent | 1.00 | 1.00 | 1.00 | 1.00 | 1.00 | 1.00 | 1.00 | 1.00 | 1.00 | 1.00 |
| smiles_ga | 1.00 | 1.00 | 1.00 | 1.00 | 1.00 | 1.00 | 1.00 | 1.00 | 1.00 | 1.00 |
| smiles_lstm_hc | 1.00 | 1.00 | 1.00 | 1.00 | 1.00 | 1.00 | 1.00 | 1.00 | 1.00 | 1.00 |
| smiles_vae | 1.00 | 1.00 | 1.00 | 1.00 | 1.00 | 1.00 | 1.00 | 1.00 | 1.00 | 1.00 |
| targetdiff | – | 0.04 | – | 1.00 | 0.99 | 0.37 | 0.88 | 0.73 | – | – |

| Model | 6GL8 | 1UWH | 7OTE | 1KKQ | 5WFD |
|---|---|---|---|---|---|
| Pocket2Mol | $0.9983 \pm 0.00$ | $0.9982 \pm 0.00$ | $0.9980 \pm 0.00$ | $0.9988 \pm 0.00$ | $0.9985 \pm 0.00$ |
| PocketFlow | $0.9984 \pm 0.00$ | $0.9964 \pm 0.00$ | $0.9983 \pm 0.00$ | $0.9987 \pm 0.00$ | $0.9972 \pm 0.00$ |
| ResGen | – | $0.9980 \pm 0.00$ | $0.9960 \pm 0.00$ | $0.9970 \pm 0.00$ | – |
| 3DSBDD | $0.9966 \pm 0.00$ | $0.9980 \pm 0.00$ | $0.9601 \pm 0.01$ | $0.9478 \pm 0.03$ | $0.9828 \pm 0.00$ |
| dst | $0.9993 \pm 0.00$ | $0.9993 \pm 0.00$ | $0.9993 \pm 0.00$ | $0.9993 \pm 0.00$ | $0.9993 \pm 0.00$ |
| graph_ga | $0.9994 \pm 0.00$ | $0.9994 \pm 0.00$ | $0.9986 \pm 0.00$ | $0.9994 \pm 0.00$ | $0.9994 \pm 0.00$ |
| jt_vae | $0.9994 \pm 0.00$ | $0.9994 \pm 0.00$ | $0.9994 \pm 0.00$ | $0.9994 \pm 0.00$ | $0.9995 \pm 0.00$ |
| mimosa | $0.9995 \pm 0.00$ | $0.9995 \pm 0.00$ | $0.9995 \pm 0.00$ | $0.9995 \pm 0.00$ | $0.9995 \pm 0.00$ |
| moldqn | $0.9936 \pm 0.00$ | $0.9942 \pm 0.00$ | $0.9935 \pm 0.00$ | $0.9932 \pm 0.00$ | $0.9921 \pm 0.00$ |
| pasithea | $0.9995 \pm 0.00$ | $0.9994 \pm 0.00$ | $0.9994 \pm 0.00$ | $0.9995 \pm 0.00$ | $0.9995 \pm 0.00$ |
| reinvent | $0.9995 \pm 0.00$ | $0.9995 \pm 0.00$ | $0.9995 \pm 0.00$ | $0.9995 \pm 0.00$ | $0.9995 \pm 0.00$ |
| smiles_ga | $0.9993 \pm 0.00$ | $0.9993 \pm 0.00$ | $0.9995 \pm 0.00$ | $0.9993 \pm 0.00$ | $0.9994 \pm 0.00$ |
| smiles_lstm_hc | $0.9996 \pm 0.00$ | $0.9996 \pm 0.00$ | $0.9996 \pm 0.00$ | $0.9996 \pm 0.00$ | $0.9996 \pm 0.00$ |
| smiles_vae | $0.9994 \pm 0.00$ | $0.9996 \pm 0.00$ | $0.9995 \pm 0.00$ | $0.9992 \pm 0.00$ | $0.9995 \pm 0.00$ |
| targetdiff | – | $0.0415 \pm 0.00$ | – | $0.9902 \pm 0.00$ | $0.8588 \pm 0.10$ |

a): targets in CrossDocking

| Model | 7W7C | 8JJL | 7D42 | 7S1S | 6AZV |
|---|---|---|---|---|---|
| Pocket2Mol | $0.9982 \pm 0.00$ | $0.9965 \pm 0.00$ | $0.9962 \pm 0.00$ | $0.9971 \pm 0.00$ | $0.9982 \pm 0.00$ |
| PocketFlow | $0.9986 \pm 0.00$ | $0.9975 \pm 0.00$ | $0.9973 \pm 0.00$ | $0.9957 \pm 0.00$ | $0.9985 \pm 0.00$ |
| ResGen | $0.9963 \pm 0.00$ | – | $0.9970 \pm 0.00$ | $0.9964 \pm 0.00$ | $0.9977 \pm 0.00$ |
| SBDD3D | $0.5402 \pm 0.28$ | $0.9697 \pm 0.01$ | $0.9881 \pm 0.00$ | $0.9678 \pm 0.02$ | $0.9918 \pm 0.00$ |
| dst | $0.9993 \pm 0.00$ | $0.9993 \pm 0.00$ | $0.9993 \pm 0.00$ | $0.9993 \pm 0.00$ | $0.9993 \pm 0.00$ |
| graph_ga | $0.9994 \pm 0.00$ | $0.9987 \pm 0.00$ | $0.9994 \pm 0.00$ | $0.9994 \pm 0.00$ | $0.9994 \pm 0.00$ |
| jt_vae | $0.9990 \pm 0.00$ | $0.9994 \pm 0.00$ | $0.9994 \pm 0.00$ | $0.9994 \pm 0.00$ | $0.9995 \pm 0.00$ |
| mimosa | $0.9995 \pm 0.00$ | $0.9995 \pm 0.00$ | $0.9995 \pm 0.00$ | $0.9995 \pm 0.00$ | $0.9995 \pm 0.00$ |
| moldqn | $0.9929 \pm 0.00$ | $0.9934 \pm 0.00$ | $0.9927 \pm 0.00$ | $0.9936 \pm 0.00$ | $0.9928 \pm 0.00$ |
| pasithea | $0.9993 \pm 0.00$ | $0.9994 \pm 0.00$ | $0.9993 \pm 0.00$ | $0.9995 \pm 0.00$ | $0.9995 \pm 0.00$ |
| reinvent | $0.9995 \pm 0.00$ | $0.9995 \pm 0.00$ | $0.9995 \pm 0.00$ | $0.9995 \pm 0.00$ | $0.9995 \pm 0.00$ |
| smiles_ga | $0.9993 \pm 0.00$ | $0.9994 \pm 0.00$ | $0.9993 \pm 0.00$ | $0.9995 \pm 0.00$ | $0.9993 \pm 0.00$ |
| smiles_lstm_hc | $0.9996 \pm 0.00$ | $0.9996 \pm 0.00$ | $0.9996 \pm 0.00$ | $0.9996 \pm 0.00$ | $0.9996 \pm 0.00$ |
| smiles_vae | $0.9995 \pm 0.00$ | $0.9995 \pm 0.00$ | $0.9996 \pm 0.00$ | $0.9996 \pm 0.00$ | $0.9994 \pm 0.00$ |
| targetdiff | $0.1287 \pm 0.16$ | $0.4119 \pm 0.33$ | $0.7323 \pm 0.00$ | – | – |

b): targets not in CrossDocking

Table 25: Top 10 RA score for each target

| Model | 6GL8 | 1UWH | 7OTE | 1KKQ | 5WFD |
|---|---|---|---|---|---|
| Pocket2Mol | $0.9939 \pm 0.00$ | $0.9929 \pm 0.00$ | $0.9910 \pm 0.01$ | $0.9888 \pm 0.01$ | $0.9967 \pm 0.00$ |
| PocketFlow | $0.9961 \pm 0.00$ | $0.9948 \pm 0.00$ | $0.9965 \pm 0.00$ | $0.9963 \pm 0.00$ | $0.9949 \pm 0.00$ |
| ResGen | $-$ | $0.9964 \pm 0.00$ | $0.9920 \pm 0.00$ | $0.9949 \pm 0.00$ | $-$ |
| 3DSBDD | $0.9885 \pm 0.01$ | $0.9663 \pm 0.03$ | $0.5018 \pm 0.33$ | $0.3784 \pm 0.30$ | $0.7572 \pm 0.20$ |
| dst | $0.9984 \pm 0.00$ | $0.9984 \pm 0.00$ | $0.9984 \pm 0.00$ | $0.9984 \pm 0.00$ | $0.9984 \pm 0.00$ |
| graph_ga | $0.9984 \pm 0.00$ | $0.9984 \pm 0.00$ | $0.9957 \pm 0.00$ | $0.9984 \pm 0.00$ | $0.9984 \pm 0.00$ |
| jt_vae | $0.9985 \pm 0.00$ | $0.9987 \pm 0.00$ | $0.9987 \pm 0.00$ | $0.9987 \pm 0.00$ | $0.9987 \pm 0.00$ |
| mimosa | $0.9991 \pm 0.00$ | $0.9991 \pm 0.00$ | $0.9991 \pm 0.00$ | $0.9991 \pm 0.00$ | $0.9991 \pm 0.00$ |
| moldqn | $0.9862 \pm 0.01$ | $0.9866 \pm 0.01$ | $0.9853 \pm 0.01$ | $0.9862 \pm 0.01$ | $0.9845 \pm 0.01$ |
| pasithea | $0.9990 \pm 0.00$ | $0.9983 \pm 0.00$ | $0.9987 \pm 0.00$ | $0.9991 \pm 0.00$ | $0.9990 \pm 0.00$ |
| reinvent | $0.9990 \pm 0.00$ | $0.9989 \pm 0.00$ | $0.9990 \pm 0.00$ | $0.9990 \pm 0.00$ | $0.9989 \pm 0.00$ |
| smiles_ga | $0.9986 \pm 0.00$ | $0.9983 \pm 0.00$ | $0.9985 \pm 0.00$ | $0.9985 \pm 0.00$ | $0.9984 \pm 0.00$ |
| smiles_lstm_hc | $0.9991 \pm 0.00$ | $0.9990 \pm 0.00$ | $0.9991 \pm 0.00$ | $0.9991 \pm 0.00$ | $0.9991 \pm 0.00$ |
| smiles_vae | $0.9990 \pm 0.00$ | $0.9991 \pm 0.00$ | $0.9991 \pm 0.00$ | $0.9982 \pm 0.00$ | $0.9990 \pm 0.00$ |
| targetdiff | $-$ | $0.0415 \pm 0.00$ | $-$ | $0.8899 \pm 0.08$ | $0.3424 \pm 0.29$ |

a): targets in CrossDocking

| Model | 7W7C | 8JJL | 7D42 | 7S1S | 6AZV |
|---|---|---|---|---|---|
| Pocket2Mol | $0.9946 \pm 0.00$ | $0.9873 \pm 0.01$ | $0.9753 \pm 0.01$ | $0.9755 \pm 0.02$ | $0.9839 \pm 0.01$ |
| PocketFlow | $0.9961 \pm 0.00$ | $0.9954 \pm 0.00$ | $0.9946 \pm 0.00$ | $0.9934 \pm 0.00$ | $0.9966 \pm 0.00$ |
| ResGen | $0.9889 \pm 0.01$ | $-$ | $0.9942 \pm 0.00$ | $0.9922 \pm 0.00$ | $0.9960 \pm 0.00$ |
| 3DSBDD | $0.1442 \pm 0.17$ | $0.2492 \pm 0.31$ | $0.8580 \pm 0.14$ | $0.3142 \pm 0.32$ | $0.9110 \pm 0.07$ |
| dst | $0.9984 \pm 0.00$ | $0.9984 \pm 0.00$ | $0.9984 \pm 0.00$ | $0.9984 \pm 0.00$ | $0.9984 \pm 0.00$ |
| graph_ga | $0.9985 \pm 0.00$ | $0.9947 \pm 0.00$ | $0.9984 \pm 0.00$ | $0.9984 \pm 0.00$ | $0.9985 \pm 0.00$ |
| jt_vae | $0.9897 \pm 0.01$ | $0.9986 \pm 0.00$ | $0.9986 \pm 0.00$ | $0.9987 \pm 0.00$ | $0.9988 \pm 0.00$ |
| mimosa | $0.9991 \pm 0.00$ | $0.9991 \pm 0.00$ | $0.9991 \pm 0.00$ | $0.9991 \pm 0.00$ | $0.9991 \pm 0.00$ |
| moldqn | $0.9846 \pm 0.01$ | $0.9859 \pm 0.01$ | $0.9842 \pm 0.01$ | $0.9860 \pm 0.01$ | $0.9861 \pm 0.01$ |
| pasithea | $0.9982 \pm 0.00$ | $0.9987 \pm 0.00$ | $0.9982 \pm 0.00$ | $0.9990 \pm 0.00$ | $0.9990 \pm 0.00$ |
| reinvent | $0.9990 \pm 0.00$ | $0.9990 \pm 0.00$ | $0.9990 \pm 0.00$ | $0.9990 \pm 0.00$ | $0.9990 \pm 0.00$ |
| smiles_ga | $0.9986 \pm 0.00$ | $0.9987 \pm 0.00$ | $0.9984 \pm 0.00$ | $0.9984 \pm 0.00$ | $0.9985 \pm 0.00$ |
| smiles_lstm_hc | $0.9991 \pm 0.00$ | $0.9991 \pm 0.00$ | $0.9991 \pm 0.00$ | $0.9991 \pm 0.00$ | $0.9991 \pm 0.00$ |
| smiles_vae | $0.9991 \pm 0.00$ | $0.9991 \pm 0.00$ | $0.9991 \pm 0.00$ | $0.9991 \pm 0.00$ | $0.9991 \pm 0.00$ |
| targetdiff | $0.1287 \pm 0.16$ | $0.4119 \pm 0.33$ | $0.7323 \pm 0.00$ | $-$ | $-$ |

b): targets not in CrossDocking

Table 26: Top 100 RA score for each target

Table 27: Posebusters evaluation results across models. The values represent the success rate of generated molecules for all receptor targets, indicating the proportion of molecules that meet the Posebusters validation criteria

| Model | All Atoms Connected | Aromatic Ring Flatness | Bond Angles | Bond Lengths | Double Bond Flatness | Inchi Convertible |
|---|---|---|---|---|---|---|
| Pocket2Mol | 0.763 | 0.990 | 0.989 | 0.988 | 0.970 | 0.559 |
| PocketFlow | 0.936 | 0.991 | 0.990 | 0.989 | 0.991 | 0.880 |
| ResGen | 0.947 | 0.988 | 0.989 | 0.989 | 0.987 | 0.913 |
| 3DSBDD | 0.459 | 0.957 | 0.943 | 0.944 | 0.922 | 0.400 |
| DST | 1.000 | 1.000 | 1.000 | 0.999 | 1.000 | 1.000 |
| graph-GA | 1.000 | 1.000 | 1.000 | 0.999 | 1.000 | 1.000 |
| JT-VAE | 1.000 | 1.000 | 1.000 | 0.998 | 1.000 | 1.000 |
| MIMOSA | 1.000 | 1.000 | 1.000 | 0.999 | 1.000 | 1.000 |
| MolDQN | 1.000 | 1.000 | 0.968 | 0.950 | 0.998 | 1.000 |
| Pasithea | 0.956 | 0.767 | 0.766 | 0.765 | 0.767 | 0.738 |
| REINVENT | 0.915 | 0.803 | 0.802 | 0.802 | 0.803 | 0.733 |
| SMILES-GA | 0.954 | 0.720 | 0.719 | 0.719 | 0.719 | 0.699 |
| SMILES-LSTM-HC | 0.930 | 0.902 | 0.900 | 0.899 | 0.902 | 0.839 |
| SMILES-VAE | 0.959 | 0.776 | 0.776 | 0.775 | 0.776 | 0.745 |
| TargetDiff | 1.000 | 1.000 | 0.758 | 0.903 | 1.000 | 0.998 |

| Model | Internal Energy | Internal Steric Clash | Min Dist To Inorg Cofactors | Min Dist To Org Cofactors | Min Dist To Protein | Min Dist To Waters |
|---|---|---|---|---|---|---|
| Pocket2Mol | 0.471 | 0.968 | 1.000 | 1.000 | 0.935 | 1.000 |
| PocketFlow | 0.724 | 0.980 | 1.000 | 1.000 | 0.998 | 1.000 |
| ResGen | 0.711 | 0.983 | 1.000 | 1.000 | 1.000 | 1.000 |
| 3DSBDD | 0.323 | 0.909 | 1.000 | 1.000 | 0.867 | 1.000 |
| DST | 0.972 | 1.000 | 1.000 | 1.000 | 0.996 | 1.000 |
| graph-GA | 0.970 | 0.999 | 1.000 | 1.000 | 0.998 | 1.000 |
| JT-VAE | 0.958 | 0.999 | 1.000 | 1.000 | 0.991 | 1.000 |
| MIMOSA | 0.968 | 1.000 | 1.000 | 1.000 | 0.997 | 1.000 |
| MolDQN | 0.947 | 0.990 | 1.000 | 1.000 | 1.000 | 1.000 |
| Pasithea | 0.724 | 0.763 | 1.000 | 1.000 | 0.996 | 1.000 |
| REINVENT | 0.717 | 0.800 | 1.000 | 1.000 | 0.998 | 1.000 |
| SMILES-GA | 0.678 | 0.714 | 1.000 | 1.000 | 0.998 | 1.000 |
| SMILES-LSTM-HC | 0.815 | 0.898 | 1.000 | 1.000 | 0.986 | 1.000 |
| SMILES-VAE | 0.735 | 0.775 | 1.000 | 1.000 | 0.997 | 1.000 |
| TargetDiff | 0.746 | 0.933 | 1.000 | 1.000 | 0.470 | 1.000 |

| Model | Mol Cond Loaded | Mol Pred Loaded | Prot-Lig Max Dist | Sanitization | Vol Overlap w/ Inorg Cof | Vol Overlap w/ Org Cof | Vol Overlap w/ Protein |
|---|---|---|---|---|---|---|---|
| Pocket2Mol | 1.000 | 1.000 | 1.000 | 0.992 | 1.000 | 1.000 | 1.000 |
| PocketFlow | 1.000 | 1.000 | 1.000 | 0.991 | 1.000 | 1.000 | 1.000 |
| ResGen | 1.000 | 1.000 | 1.000 | 0.989 | 1.000 | 1.000 | 1.000 |
| 3DSBDD | 1.000 | 1.000 | 1.000 | 0.957 | 1.000 | 1.000 | 1.000 |
| DST | 1.000 | 1.000 | 1.000 | 1.000 | 1.000 | 1.000 | 1.000 |
| graph-GA | 1.000 | 1.000 | 1.000 | 1.000 | 1.000 | 1.000 | 1.000 |
| JT-VAE | 1.000 | 1.000 | 1.000 | 1.000 | 1.000 | 1.000 | 1.000 |
| MIMOSA | 1.000 | 1.000 | 1.000 | 1.000 | 1.000 | 1.000 | 1.000 |
| MolDQN | 1.000 | 1.000 | 1.000 | 1.000 | 1.000 | 1.000 | 0.996 |
| Pasithea | 1.000 | 1.000 | 1.000 | 0.767 | 1.000 | 1.000 | 1.000 |
| REINVENT | 1.000 | 1.000 | 1.000 | 0.803 | 1.000 | 1.000 | 1.000 |
| SMILES-GA | 1.000 | 1.000 | 1.000 | 0.720 | 1.000 | 1.000 | 1.000 |
| SMILES-LSTM-HC | 1.000 | 1.000 | 1.000 | 0.902 | 1.000 | 1.000 | 1.000 |
| SMILES-VAE | 1.000 | 1.000 | 1.000 | 0.776 | 1.000 | 1.000 | 1.000 |
| TargetDiff | 1.000 | 1.000 | 1.000 | 0.998 | 1.000 | 1.000 | 0.903 |

Table 28: Clash statistics across different models. An empty item (−) means the model either did not generate molecules for that target or the evaluation failed. UQ stands for Upper Quartile.

| Model | 6GL8 | | | | 1UWH | | | | 7OTE | | | | 1KKQ | | | | 5WFD | | | |
|---|---|---|---|---|---|---|---|---|---|---|---|---|---|---|---|---|---|---|---|---|
| | Min | Max | Mean | UQ | Min | Max | Mean | UQ | Min | Max | Mean | UQ | Min | Max | Mean | UQ | Min | Max | Mean | UQ |
| DST | 0 | 14 | 3.14 | 4 | 0 | 23 | 5.54 | 7 | 0 | 19 | 5.29 | 7 | 0 | 17 | 4.14 | 6 | 0 | 11 | 3.44 | 5 |
| JT-VAE | 0 | 14 | 3.56 | 5 | 0 | 60 | 6.28 | 8 | 0 | 30 | 6.44 | 9 | 0 | 21 | 4.96 | 6 | 0 | 16 | 4.00 | 5 |
| MIMOSA | 0 | 11 | 3.15 | 4 | 0 | 21 | 5.35 | 7 | 0 | 22 | 5.62 | 8 | 0 | 16 | 4.09 | 6 | 0 | 12 | 3.46 | 5 |
| MolDQN | 0 | 13 | 2.74 | 4 | 0 | 13 | 2.78 | 4 | 0 | 19 | 3.14 | 4 | 0 | 9 | 2.39 | 3 | 0 | 13 | 3.03 | 4 |
| graph-GA | 0 | 15 | 2.34 | 3 | 0 | 25 | 3.34 | 5 | 0 | 20 | 4.88 | 7 | 0 | 13 | 2.59 | 4 | 0 | 11 | 2.24 | 3 |
| Pasithea | 0 | 12 | 3.02 | 4 | 0 | 26 | 5.58 | 7 | 0 | 23 | 5.51 | 7 | 0 | 20 | 4.17 | 6 | 0 | 11 | 3.41 | 5 |
| REINVENT | 0 | 13 | 3.09 | 4 | 0 | 21 | 4.42 | 6 | 0 | 20 | 4.42 | 6 | 0 | 18 | 4.02 | 5 | 0 | 14 | 3.32 | 4 |
| SMILES-GA | 0 | 13 | 3.34 | 4 | 0 | 18 | 5.15 | 7 | 0 | 21 | 4.89 | 7 | 0 | 16 | 4.12 | 5 | 0 | 10 | 3.57 | 5 |
| SMILES-LSTM-HC | 0 | 14 | 3.56 | 5 | 0 | 38 | 6.38 | 8 | 0 | 29 | 6.48 | 8 | 0 | 27 | 5.02 | 7 | 0 | 16 | 4.04 | 5 |
| SMILES-VAE | 0 | 9 | 2.93 | 4 | 0 | 25 | 5.45 | 7 | 0 | 22 | 5.41 | 7 | 0 | 17 | 4.01 | 5 | 0 | 10 | 3.42 | 5 |
| 3DSBDD | 0 | 11 | 3.31 | 5 | 0 | 15 | 5.06 | 7 | 0 | 28 | 8.85 | 12 | 0 | 15 | 5.79 | 7 | − | − | − | − |
| Pocket2Mol | 0 | 15 | 4.14 | 5 | 0 | 49 | 9.09 | 12 | 0 | 34 | 8.28 | 11 | 0 | 27 | 7.73 | 10 | 0 | 15 | 4.30 | 6 |
| PocketFlow | 0 | 14 | 2.72 | 4 | 0 | 19 | 2.73 | 4 | 0 | 19 | 3.26 | 5 | 0 | 17 | 3.11 | 5 | 0 | 18 | 3.74 | 5 |
| ResGen | − | − | − | − | 0 | 9 | 2.32 | 3 | 0 | 8 | 1.61 | 2 | 0 | 16 | 5.34 | 7 | − | − | − | − |
| TargetDiff | − | − | − | − | 88 | 88 | 88.00 | 88 | − | − | − | − | 3 | 115 | 30.43 | 40 | 19 | 93 | 48.75 | 61 |

| Model | 7W7C | | | | 8JJL | | | | 7D42 | | | | 7S1S | | | | 6AZV | | | |
|---|---|---|---|---|---|---|---|---|---|---|---|---|---|---|---|---|---|---|---|---|
| | Min | Max | Mean | UQ | Min | Max | Mean | UQ | Min | Max | Mean | UQ | Min | Max | Mean | UQ | Min | Max | Mean | UQ |
| DST | 0 | 25 | 4.13 | 6 | 0 | 39 | 10.26 | 13 | 0 | 28 | 4.86 | 6 | 0 | 21 | 5.49 | 7 | 0 | 19 | 4.87 | 6 |
| JT-VAE | 0 | 47 | 5.69 | 7 | 0 | 74 | 13.87 | 18 | 0 | 66 | 7.15 | 9 | 0 | 25 | 6.78 | 9 | 0 | 21 | 5.37 | 7 |
| MIMOSA | 0 | 27 | 4.23 | 6 | 0 | 47 | 10.33 | 13 | 0 | 38 | 4.68 | 6 | 0 | 25 | 5.46 | 7 | 0 | 18 | 4.83 | 6 |
| MolDQN | 0 | 9 | 1.77 | 3 | 0 | 20 | 4.03 | 5 | 0 | 13 | 2.26 | 3 | 0 | 15 | 3.46 | 5 | 0 | 12 | 2.33 | 3 |
| graph-GA | 0 | 29 | 2.57 | 3 | 0 | 51 | 9.04 | 13 | 0 | 33 | 2.91 | 3 | 0 | 16 | 3.35 | 4 | 0 | 19 | 2.73 | 4 |
| Pasithea | 0 | 33 | 4.14 | 5 | 0 | 52 | 10.28 | 14 | 0 | 30 | 5.01 | 6 | 0 | 20 | 5.26 | 7 | 0 | 22 | 4.62 | 6 |
| REINVENT | 0 | 32 | 4.18 | 6 | 0 | 36 | 7.81 | 10 | 0 | 36 | 4.91 | 6 | 0 | 27 | 5.52 | 7 | 0 | 21 | 4.68 | 6 |
| SMILES-GA | 0 | 25 | 4.46 | 6 | 0 | 31 | 9.82 | 13 | 0 | 22 | 4.33 | 6 | 0 | 18 | 5.25 | 7 | 0 | 17 | 4.79 | 7 |
| SMILES-LSTM-HC | 0 | 69 | 6.99 | 9 | 0 | 98 | 15.51 | 19 | 0 | 88 | 8.81 | 11 | 0 | 28 | 6.42 | 8 | 0 | 20 | 5.22 | 7 |
| SMILES-VAE | 0 | 30 | 4.14 | 6 | 0 | 40 | 10.06 | 13 | 0 | 26 | 4.66 | 6 | 0 | 17 | 5.27 | 7 | 0 | 19 | 4.82 | 6 |
| 3DSBDD | − | − | − | − | − | − | − | − | 0 | 12 | 5.19 | 7 | 0 | 20 | 7.20 | 9 | 0 | 44 | 8.10 | 11 |
| Pocket2Mol | 0 | 64 | 8.67 | 9 | 0 | 103 | 20.42 | 26 | 0 | 84 | 23.82 | 33 | 0 | 34 | 7.48 | 10 | 0 | 21 | 6.56 | 9 |
| PocketFlow | 0 | 26 | 5.75 | 9 | 0 | 48 | 8.64 | 13 | 0 | 64 | 8.09 | 11 | 0 | 23 | 4.82 | 6 | 0 | 20 | 4.33 | 6 |
| ResGen | 0 | 6 | 1.64 | 2 | − | − | − | − | 0 | 14 | 3.41 | 5 | 0 | 16 | 4.70 | 6 | 0 | 20 | 4.09 | 6 |
| TargetDiff | 140 | 173 | 160.00 | 170 | 61 | 176 | 112.90 | 130 | 51 | 51 | 51.00 | 51 | − | − | − | − | − | − | − | − |

