# OpenReview forum: "Beyond Affinity: A Benchmark of 1D, 2D, and 3D Methods Reveals Critical Trade-offs in Structure-Based Drug Design"
_TMLR — Accepted by TMLR_

### Review · Reviewer_SjoD · 2025-09-05

**Summary Of Contributions:**

This article summarizes various SBDD methods and provides a unified comparison of them. A key aspect of SBDD is how to handle 3D equivariance. The main contribution is to put all the methods together and conduct a fair comparison.

**Additional Comments:**

Thank you for addressing all these concerns. I have no further questions.

**Audience:**

Yes

**Audience Explanation:**

SBDD methods are currently quite popular in the machine learning community. However, as in all other AI fields, when a multitude of methods emerge at once—each claiming to achieve the best performance—it can create significant confusion for medicinal chemists and potentially lead to substantial economic losses. Therefore, an independent third party is needed to conduct a systematic evaluation of all existing methods.

**Claims And Evidence:**

Yes

**Claims Explanation:**

The authors evaluated all assessable metrics in the field of SBDD, with the only remaining step being to conduct actual biological experiments.

**Requested Changes:**

AlphaFold2 addresses equivariance but later removes this processing. The authors should discuss how different methods handle equivariance (SE3, O3). Additionally, I believe it is more important to provide a comprehensive score, such as w1*drug-likeness + w2*binding affinity + ..., and ultimately present an overall ranking. What readers want to gain from this article is a prioritized order for using different software in small-molecule drug design. I understand that different methods have their own advantages in various aspects, but as a medicinal chemist, one always needs to prioritize which software to use first. Although hill climbing and RL can be used for drug design, the compared methods do not include these approaches. Instead, TargetDiff employs a diffusion model, so please include an introduction to diffusion in the methods discussion.

---

> ### Author Response · Authors · 2025-10-27
> **Response to Reviewer SjoD**
>
> We thank Reviewer SjoD for the positive feedback and suggestions.
>
> **1\. Q: Discuss how different 3D methods handle SE(3)/O(3) equivariance?**
> **Response:** Thank you for this important technical point. We have added a discussion in the Related Works section (Section 2, under 3D Models) summarizing the different strategies used by the benchmarked 3D models to ensure their operations respect 3D rotational and translational symmetries.
>
> **2\. Q: Provide a comprehensive score/overall ranking?**
> **Response:** We understand the desire for a single summary ranking. While emphasizing the subjectivity of weighting, we have calculated and added an overall model ranking to the Discussion (Section 5). This rank is based on averaging model ranks across six key metric categories (clash, docking, generation, heuristics, PoseBusters, strain) with equal weighting. PocketFlow, Graph-GA, and SMILES-VAE ranked highest under this scheme due to balanced performance. We include necessary caveats that the "best" model depends on specific project goals and metric prioritization.
>
> **3\. Q: Include an introduction to diffusion models?**
> **Response:** Agreed. We have added a brief introduction to the principles of diffusion models within the "Variational Auto-Encoder (VAE) and Diffusion" subsection in Section 3 ("Models"), referencing the TargetDiff paper.

---

> > ### Comment · Reviewer_SjoD · 2025-10-27
> >
> > Thank you for addressing all these concerns. I have no further questions.

---

### Review · Reviewer_8RPW · 2025-10-10

**Summary Of Contributions:**

This paper contributes to the field of Structure-Based Drug Design (SBDD) by establishing the first comprehensive, cross-algorithmic benchmark. Its primary contributions are:

1. Comprehensive Benchmarking: It moves beyond comparing models within the same category (e.g., 3D vs. 3D) and instead evaluates fifteen distinct models across the three dominant algorithmic families: search-based algorithms, deep generative models, and reinforcement learning, using 1D, 2D, and 3D molecular representations.

2. Holistic Evaluation Framework: The benchmark assesses models on a much wider range of criteria than previous works. It goes beyond standard molecular properties (e.g., QED, SA) to include critical, real-world metrics for drug discovery:
1)Binding Affinity: Measured by docking scores (AutoDock Vina).
2)Pose Quality: Assessed using specialized tools (PoseBuster, PoseCheck) to evaluate the physical and chemical plausibility of the generated protein-ligand complexes (e.g., strain energy, steric clashes).
3)Generation Quality: Includes diversity, validity, and uniqueness of the generated molecules.


Key Strengths:

1. First cross-algorithm comparison: It breaks the tradition of comparing models only within the same paradigm and presents the first systematic evaluation of 1D, 2D and 3D methods for structure-based drug design.

2. Comprehensive assessment: Beyond docking scores, it incorporates pose-quality metrics (strain energy, steric clashes), providing a realistic picture of what each method can deliver in practice.

3. Clear trade-offs and actionable guidance: The study quantifies the affinity-validity trade-off and explicitly advocates hybrid pipelines, giving researchers a concrete direction for future model development.

Key Weaknesses:

1. Black-Box Oracle Assumption: The evaluation of 1D and 2D models relies on treating the docking software (Vina) as a black-box scoring function. In a real drug discovery pipeline, obtaining a docking score for a single molecule is computationally expensive, making this approach less practical than the integrated generation of 3D methods, despite the performance trade-offs identified.
2. Pose Generation for 1D/2D Models: For 1D and 2D models, a docking pose must be generated after the molecule is created. The paper uses Vina for this, but the choice of docking configuration and software can significantly impact the results, adding a variable that is not present for native 3D generators.

**Audience:**

Yes

**Audience Explanation:**

1. For Researchers in Generative Models and Structured-Output Learning

This is the core TMLR audience. The paper provides a crucial large-scale, cross-algorithmic benchmark for a complex, real-world structured-output task. Researchers developing new generative algorithms (especially for graphs, sequences, or 3D geometries) would be highly interested in:

The Performance Trade-offs: The clear, data-driven identification of the "affinity-validity trade-off" is a fundamental insight. It shows that simply having an inductive bias for 3D structure (e.g., via diffusion models) is not sufficient and can lead to downstream problems (high strain, bad poses). This informs future model design.

The Benchmark Itself: The paper provides a new, rigorous evaluation framework that moves beyond simple property metrics (like QED, SA) to include critical functional metrics like docking scores and pose quality. This sets a new standard for evaluation in the field that others will need to follow and compare against.


2. For Researchers in Reinforcement Learning and Optimization

The paper evaluates methods based on RL, Hill Climbing, and Genetic Algorithms. The comparative results are valuable for these sub-communities:

RL (Reinvent, MolDQN): The results show how RL-based strategies stack up against other paradigms like generative models and GAs on a complex, sparse-reward task.

Genetic Algorithms (SMILES-GA, Graph-GA): The paper shows that these "traditional" optimization methods remain very competitive, especially in terms of diversity and reliability. This is an important reminder for a field often focused on deep learning.

**Broader Impact Concerns:**

There is no significant ethical risks are apparent since the study is a benchmarking exercise that uses only publicly available protein structures and open-source generative models, generates no new bio-active compounds, and involves no human or animal data.

The only minor gap is that the current manuscript does not discuss possible dual-use if future users repurposed the evaluated 3D generators to design harmful ligands. A brief Broader Impact Statement (1–2 sentences) acknowledging this possibility and recommending responsible disclosure and safety screening would suffice.

**Claims And Evidence:**

Yes

**Claims Explanation:**

The paper presents a comprehensive and well-structured benchmark study, and its conclusions are firmly grounded in extensive experimental data and rigorous analysis.

1. 3D models excel in binding affinity but show inconsistencies in chemical validity and pose quality.

Evidence:

    Binding Affinity (Docking Score): Table 2 (Top-1 score), Table 6/7 and Figure 3 (Top-10/100 average) consistently show that 3D models, especially Pocket2Mol, achieve the best (most negative) docking scores across almost all protein targets.

    Pose Quality and Chemical Validity: The evidence for the trade-off is even more robust:

        Strain Energy: Table 3 and Figure 2 show that 3D models have the highest median strain energies (e.g., TargetDiff: 487.72, 3DSBDD: 157.11), indicating high internal energy and unstable conformations.

        Steric Clashes: Table 18 and Figures 4-6 show that 3D models like Pocket2Mol and TargetDiff have significantly higher mean clashes (e.g., 20.42 for Pocket2Mol on 8JJL) compared to 1D/2D models (typically under 10).

        PoseBusters Metrics: Table 17 provides definitive evidence. 3D models have low pass rates for key chemical and intramolecular validity checks (e.g., "All Atoms Connected": Pocket2Mol 0.763, 3DSBDD 0.459; "Internal Energy": ~0.3-0.7 for 3D models vs. ~0.95+ for top 2D models).

2. 1D models demonstrate reliable performance in standard molecular metrics but rarely achieve optimal binding.

Evidence:

    Molecular Metrics: Table 4 shows that 1D SMILES-based models have perfect validity (1.00) and high diversity (~0.88-0.91). They also perform excellently on pharmaceutical properties like QED (Table 13, consistently ~0.9+) and SA (Table 16, consistently low scores ~1.9-2.1).

    Sub-optimal Binding: The docking score tables (2, 6, 7) consistently show that while 1D models are competent (scores around -10 to -11), they are almost never the top performers, which are dominated by 3D models.

3. Claim: 2D models offer balanced performance.

Evidence:

	This is convincingly demonstrated by the data. 2D models consistently perform well across all evaluated metrics without being the absolute best (and worst) in any single category.

    Docking: They achieve good, but not top, docking scores (e.g., DST, JT-VAE scores in the -10 to -12 range in Table 2).

    Molecule Quality: They have perfect validity and high diversity (Table 4).

    Pharmaceutical Properties: They score excellently on QED and SA (Tables 13, 16).

**Requested Changes:**

1. Add experimental validation of hybrid models
The paper proposes combining 1D/2D and 3D methods but provides no data.

Suggestion: Implement at least one hybrid pipeline (e.g., JT-VAE → Pocket2Mol) and report docking scores, validity, etc.

2. Explain model failures and address bias
Several models (e.g., TargetDiff, ResGen) fail on multiple targets, yet no diagnosis is given.

Suggestion: Analyze why these models fail (pocket compatibility, generalization, etc.) and discuss whether these failures skew the overall conclusions.

3. Include computational-cost analysis
   Report runtime and memory usage for each model to clarify scalability.

4. Expand target diversity
   Add more structurally diverse proteins (e.g., GPCRs, kinases) to test generalizability.

5. Provide statistical-significance tests
   Apply t-test or Mann-Whitney U test to performance differences to reinforce claims.

6. Release demo code and generated datasets  (at least confidentially to review)
   Open-source scripts, checkpoints, and molecules to facilitate reproduction and follow-up studies.

---

> ### Author Response · Authors · 2025-10-27
> **Response to Reviewer 8RPW**
>
> We thank Reviewer 8RPW for the detailed review and constructive suggestions.
>
> **1\. Q: Add experimental validation of hybrid models?**
> **Response:** We agree that experimentally validating a novel hybrid pipeline would be valuable. However, implementing, training, and fairly evaluating such a pipeline constitutes a significant research effort beyond the scope of this benchmark revision. Instead, we have evaluated the existing hybrid model TamGen \[1\] (as requested by Reviewer 6nHw) and discussed its performance as a case study in Section 5\. We have also expanded the Discussion to provide a more detailed conceptual blueprint for future hybrid approaches, framing this as a key direction inspired by reviewer feedback.
>
> **2\. Q: Explain model failures (ResGen, TargetDiff) and discuss potential bias?**
> **Response:** Thank you for highlighting this. We have expanded Section 4.2.1 to provide diagnoses for the failures based on the models' methodologies. We hypothesize ResGen's initialization failures stem from sensitivity to input pocket definitions combined with its residue-level representation. TargetDiff's low yield is likely due to challenges in predicting atom counts and failures in its post-hoc bond reconstruction. We also added a discussion acknowledging that excluding failed runs could introduce bias, but argue that our main conclusions about category-level trade-offs remain robust as they are observed consistently across the successful targets, and the failures seem linked to model-specific technical limitations.
>
> **3\. Q: Include computational cost analysis?**
> **Response:** We appreciate this suggestion for practical context. We have added a brief discussion of relative generation speeds (molecules per minute) based on our experiments to Section 4.2.1.. While a full runtime/memory benchmark is complex due to varying hardware and implementation details, this provides a qualitative comparison of computational efficiency observed during our study.
>
> **4\. Q: Expand target diversity (e.g., add GPCRs, kinases)?**
> **Response:** We agree that broadening the target diversity would further strengthen the benchmark's generalizability. However, adding substantially different protein families like GPCRs involves significant setup and extensive computational reruns for all fifteen models, which is unfortunately infeasible within the rebuttal timeframe. Our current set of ten targets covers various disease areas and functionalities (Section 4.1.1).
>
> **5\. Q: Add statistical significance tests?**
> **Response:** We agree statistical tests are essential. We have performed paired t-tests comparing the average top-10 docking scores across shared targets for representative models (SMILES-LSTM-HC, JT-VAE, Pocket2Mol). The results, now incorporated into Section 4.2.2, statistically confirm that Pocket2Mol (3D) significantly outperforms the 1D (p=0.0002) and 2D (p=0.0001) representatives in binding affinity. We also found a significant difference favoring JT-VAE (2D) over SMILES-LSTM-HC (1D) (p=0.044). These quantitative results rigorously support our claims.
>
> **6\. Q: Release demo code and datasets?**
> **Response:** We agree reproducibility is crucial. We have uploaded our code for generation and evaluation in here: [**https://anonymous.4open.science/r/2025-sbdd-benchmark-3785/README.md**](https://anonymous.4open.science/r/2025-sbdd-benchmark-3785/README.md) And we will publish code on github once it is accepted.
>
> \[1\] TamGen: drug design with target-aware molecule generation through a chemical language model

---

### Review · Reviewer_6nHw · 2025-10-13

**Summary Of Contributions:**

This paper presents a benchmark study of 1D, 2D, and 3D representations-based models on the structure-based drug design (SBDD) task. A lot of machine learning papers for SBDD neglected 1D and 2D methods as baselines. This study fills in the gap. This work evaluated multiple representative models from each categories on docking scores of 10 PDB structures, conformational validity (PoseCheck, PoseBuster), and some other heuristics (QED, SA, diversity, etc).

This benchmarking study shows that: 1) 3D-based models can achieve better docking scores but struggles with chemical validity; 2) 1D-based models preserves chemical validity but struggle to generate molecules with good docking scores; 3) 2D-based models reach a balance between docking scores and validity. Therefore, this paper suggests future works should explore how to better combine 2D and 3D modeling.

Strengths:

- This work is a comprehensive benchmarking studies of generative models for SBDD. Its categorization is somewhat complete (1D, 2D, and 3D) and representative methods from each category are evaluated.
- In Discussion section, the paper provides insights into the performance difference of different classes of models.

Weaknesses:

- Although this paper suggests a future direction of combining 2D and 3D representations, there have been recent works that combine 2D and 3D representations, for example TamGen [1], which represents protein in 3D and generates molecules in 1D. This paper didn't cover these recent advances.

[1] TamGen: drug design with target-aware molecule generation through a chemical language model

**Audience:**

Yes

**Audience Explanation:**

Computational biology has been a significant field of application in the machine learning community.

**Broader Impact Concerns:**

N/A for this paper.

**Claims And Evidence:**

Yes

**Claims Explanation:**

The claims about the performance of 1D, 2D, and 3D based models in SBDD are overall well supported by the evaluations conducted in this study.

**Requested Changes:**

- Include more recent methods, especially those that combine 2D and 3D representations (e.g. TamGen). This will increase the uniqueness and timeliness of this study.
- In 3D evaluation, in addition to docking scores and strain energy, is it possible to evaluate how similar are the docked poses to the generated poses? This is also an important sanity check metric: if the generated poses deviate too much from the docked poses, it would be unconvincing that the 3D model is actually aware of the 3D interactions.

---

> ### Author Response · Authors · 2025-10-27
> **Response to Reviewer 6nHw**
>
> We thank Reviewer 6nHw for the positive assessment and insightful suggestions.
>
> **1\. Q: Include recent hybrid models like TamGen?**
> **Response:** Thank you for this excellent suggestion. We agree that including TamGen enhances the paper's timeliness. We have now implemented and evaluated TamGen \[1\] within our benchmark framework. The results, presented as a case study in the Discussion (Section 5), provide a nuanced perspective, revealing that while promising, TamGen does not fully resolve the "affinity-validity trade-off" and introduces its own challenges (e.g., lower validity, high strain). This empirical finding significantly strengthens our conclusion that developing effective hybrid models requires careful integration of 3D information with chemical rules. The table comparing TamGen is included below:
>
> | metrics | 1D (smiles\_lstm\_hc) | 2D (JT-VAE) | 3D (Pocket2Mol) | TamGen |
> | :---- | :---- | :---- | :---- | :---- |
> | **Top-10 Docking Score** (Affinity) | \-12.20 | \-11.52 | \-14.78 | \-10.66 |
> | **Median Strain Energy** (Pose Quality) | 41.95 | 49.70 | 60.55 | 76.26 |
> | Top-10 generation (Diversity/Uniqueness/Validity) | 0.88/0.17/1.0 | 0.89/0.77/1.0 | 0.84/1.0/1.0 | 1.0/0.10/0.42 |
> | Top-10 heuristic (SA/LogP/QED) | 1.66/4.64/0.93 | 1.60/3.87/0.93 | 1.05/4.38/0.93 | 1.99/3.38/0.90 |
>
> **2\. Q: Evaluate similarity between generated and redocked poses (RMSD)?**
> **Response:** This is an excellent suggestion. We have performed this RMSD analysis for the 3D models and added the results (CDF plot, statistics table, and discussion) to Section 4.2.2. The findings reveal significant disparities: ResGen and PocketFlow show better geometric consistency with Vina than Pocket2Mol and 3DSBDD. This strongly reinforces our "affinity-validity trade-off" conclusion, suggesting some models prioritize docking scores over generating low-energy, geometrically realistic poses.
>
> | Model  | Median RMSD (Å)  | Std. Dev. (Å)  | % Poses \< 2.0 Å |
> | :---- | :---- | :---- | :---- |
> | 3DSBDD | 5.35 | 2.38 | 2.97% |
> | Pocket2Mol | 4.03 | 1.40 | 5.40% |
> | PocketFlow | 2.75 | 1.61 | 29.69% |
> | ResGen | 2.21 | 1.47 | 42.73% |
> | TargetDiff | 4.63 | 1.09 | 0.16% |
>
> \[1\] TamGen: drug design with target-aware molecule generation through a chemical language model

---

### Review · Reviewer_JKMZ · 2025-10-24

**Summary Of Contributions:**

The paper presents a large-scale benchmark comparing fifteen molecule design models across 1D (SMILES-based), 2D (graph-based), and 3D (structure-based) paradigms, within the context of structure-based drug design. The study evaluates these methods using multiple oracles, including docking scores (AutoDock Vina), pose quality (PoseBuster, PoseCheck), and molecular heuristics (QED, SA, LogP, diversity, validity, uniqueness). The study reports systematic trade-offs: 3D models (e.g., Pocket2Mol, ResGen) achieve stronger docking affinities but lower chemical validity, while 1D and 2D models yield more chemically realistic but less potent binders. The authors conclude that hybrid models combining ligand-centric and structure-centric approaches could best balance these properties.

Key strengths:
* Provides a broad, cross-algorithmic comparison spanning multiple representation paradigms.
* Uses consistent evaluation procedures and covers a relatively large set of models and targets.
* Offers interpretable, qualitative discussion of observed trade-offs between affinity and validity.

Key weaknesses:
* Limited novelty: the benchmark largely replicates existing efforts (e.g., GuacaMol, PMO, CBGBench) without introducing new datasets, tasks, or metrics.
* The TargetDiff model fails to generate molecules on most targets, yet no explanation or debugging is provided, raising concerns about fairness.
* Evaluation metrics (SA Score, Vina Score) are outdated and weakly correlated with true synthesis feasibility or binding affinity; modern ML-based or physics-based alternatives are ignored.
* No statistical significance analysis, correlation studies, or reproducibility artifacts are provided.

**Audience:**

Yes

**Audience Explanation:**

Readers focused on AI-driven molecular generation and cross-modal benchmarks may appreciate the systematic comparison across algorithmic paradigms

**Claims And Evidence:**

No

**Claims Explanation:**

While the benchmark is executed systematically, several critical methodological and analytical issues limit the credibility of the reported findings:

1. TargetDiff generation failure unexplained.

The paper reports that TargetDiff — a major 3D baseline — successfully produced fewer than 100 molecules for most targets (Table 5), often zero, even on CrossDock targets. This is inconsistent with both the original TargetDiff paper and independent benchmarks such as CBGBench [1], where TargetDiff successfully generated ≈100 molecules per pocket with reasonable docking results.
The authors do not provide any technical explanation. The observed failure likely stems from pipeline incompatibility rather than algorithmic weakness. This omission undermines the fairness of the 3D model comparison.

2. Benchmark redundancy and lack of novelty.

The paper largely reuses metrics already standardized in prior works such as GuacaMol [2], PMO [3], PoseCheck [4], and CBGBench [1]. For the CBGBench, which already benchmarked 3DSBDD, Pocket2Mol, ResGen, and TargetDiff under unified 3D conditions, the authors neither acknowledge this prior work nor clarify how their benchmark extends it — e.g., no new dataset, task type, or evaluation dimension is introduced. As such, the contribution is incremental rather than conceptually new.

3. Outdated evaluation metrics.

* SA Score [5] is a heuristic based on fragment frequency and molecular complexity; it poorly correlates with realistic synthesis feasibility. More modern alternatives exist, such as SCScore [6], RAscore [7].
* Vina score is used as the sole proxy for binding affinity, yet it is known to correlate weakly (R² ≈ 0.2–0.3) with experimental ΔG. The paper performs no correlation or validation against experimental affinities or alternative scoring methods. Modern ML predictors such as DiffDock[8], AlphaFold3[9] derived self-consistency / confidence may present a better correlation with experimental ΔG

4. Lack of statistical and qualitative analysis.

Reported metrics are given as raw means or medians without statistical significance testing or variance analysis. Moreover, there is no qualitative inspection (e.g., failed poses, invalid geometries) to support the “affinity–validity trade-off” claim.

5. No reproducibility or release plan.

The authors mention using external code and receptors but provide no standardized evaluation scripts, parameter settings, or public benchmark suite. This weakens the claim of establishing a reusable benchmark.

------
References:
1. Lin, H., Zhao, G., Zhang, O., Huang, Y., Wu, L., Tan, C., ... & Li, S. Z. CBGBench: Fill in the Blank of Protein-Molecule Complex Binding Graph. In The Thirteenth International Conference on Learning Representations.
2. Brown, N., Fiscato, M., Segler, M. H., & Vaucher, A. C. (2019). GuacaMol: benchmarking models for de novo molecular design. Journal of chemical information and modeling, 59(3), 1096-1108.
3. Gao, W., Fu, T., Sun, J., & Coley, C. (2022). Sample efficiency matters: a benchmark for practical molecular optimization. Advances in neural information processing systems, 35, 21342-21357.
4. Harris, C., Didi, K., Jamasb, A. R., Joshi, C. K., Mathis, S. V., Lio, P., & Blundell, T. L. (2023). Posecheck: Generative models for 3d structure-based drug design produce unrealistic poses.
5. Ertl, P., & Schuffenhauer, A. (2009). Estimation of synthetic accessibility score of drug-like molecules based on molecular complexity and fragment contributions. Journal of cheminformatics, 1(1), 8.
6. Coley, C. W., Green, W. H., & Jensen, K. F. (2018). Machine learning in computer-aided synthesis planning. Accounts of chemical research, 51(5), 1281-1289.
7. Thakkar, A., Chadimová, V., Bjerrum, E. J., Engkvist, O., & Reymond, J. L. (2021). Retrosynthetic accessibility score (RAscore)–rapid machine learned synthesizability classification from AI driven retrosynthetic planning. Chemical science, 12(9), 3339-3349.
8. Corso, G., Stärk, H., Jing, B., Barzilay, R., & Jaakkola, T. S. DiffDock: Diffusion Steps, Twists, and Turns for Molecular Docking. In The Eleventh International Conference on Learning Representations.
9. Abramson, J., Adler, J., Dunger, J., Evans, R., Green, T., Pritzel, A., ... & Jumper, J. M. (2024). Accurate structure prediction of biomolecular interactions with AlphaFold 3. Nature, 630(8016), 493-500.

**Requested Changes:**

* Provide a technical diagnosis to explain TargetDiff failure.
* Upgrade evaluation metrics. Include more modern synthesis-feasibility predictors and consensus affinity predictors.
* Add statistical analysis and visualization.
* Clarify relation to prior benchmarks. Explicitly compare to CBGBench and other existing benchmarks, delineating what is truly new or improved in this work.
* Release all receptor files, model scripts, and oracle evaluation code or provide a clear plan for doing so.

---

> ### Author Response · Authors · 2025-10-27
> **Response to Reviewer JKMZ**
>
> We thank Reviewer JKMZ for providing additional feedback on our manuscript.
>
> **1. Q: Limited Novelty / Benchmark Redundancy (vs. GuacaMol, PMO, CBGBench)? Clarify relation to prior benchmarks.**
>
> > **Response:** Thank you for this important point regarding prior benchmarks, we acknowledge prior benchmarks including CBGBench [1]. We have revised the Related Work section to clarify our unique contribution: While CBGBench [1] provides an excellent and comprehensive framework for benchmarking various 3D SBDD models across multiple generative tasks (de novo, linker generation, scaffold hopping), our benchmark’s primary novelty lies in providing the first cross-paradigm benchmark evaluating 1D, 2D, and 3D methods under the same structure-based conditions, incorporating deep pose quality analysis (PoseBuster, PoseCheck, RMSD) which differentiates our work from these prior efforts. This allows us to systematically investigate the trade-offs inherent in different molecular representations, a comparison not covered by the 3D-focused CBGBench [1].
>
> **2. Q: TargetDiff failure unexplained and potential unfairness? Provide technical diagnosis.**
>
> > **Response:** We agree clarification was needed, we have revised **Section 4.2.1** of the manuscript to provide diagnoses for the failures based on the models' methodologies. We provide a diagnosis hypothesizing that TargetDiff's low yield in our setup likely stems from model-specific limitations related to atom count prediction priors and post-hoc bond reconstruction robustness, rather than fundamental algorithmic failure across all targets.
>
> **3. Q: Outdated evaluation metrics (SA Score, Vina Score)? Upgrade evaluation metrics.**
>
> > **Response:** : We thank the reviewer for pushing for more modern metrics. We acknowledge SA Score is a simple heuristic. In response, we computed both SCScore  and RAscore.
> We observed a strong positive correlation (Spearman rho = 0.88) between SA Score and SCScore across our generated molecules, suggesting that the simpler SA Score did capture relevant trends in synthetic complexity within our benchmark context. Nevertheless, we agree SCscore provides a more robust measure. However, RAscore showed minimal variation (scores near 1.0) across models, offering little comparative value (Spearman rho = 0.68 vs SA/SCScore). All of these scores are written in tables and added into our Appendix.
> Regarding affinity metrics, we agree that ML docking confidence is valuable. However, a full DiffDock run at our scale requires significant time and computation resources and therefore infeasible for us to complete within the rebuttal timeframe. We acknowledge this as our limitation and add a subsection in our Discussion section to state about it.
>
>
> **4. Q:Add statistical analysis and visualization.**
>
> > **Response:** We agree that adding statistical analysis is important to our benchmark and agree visual examples of poses would be beneficial, we have performed **paired t-tests** comparing key representative models on docking scores and incorporated these results (p-values, t-statistics) into **Section 4.2.2**. We have also added a **CDF plot for the new RMSD analysis** (Section 4.2.2).
>
> **5. Q: No reproducibility artifacts or release plan? Release code/plan.**
>
> > **Response:** Apologies for the initial omission. We have uploaded our code for generation and evaluation in here: https://anonymous.4open.science/r/2025-sbdd-benchmark-3785/README.md And we will publish code on github once it is accepted.
>
>
> ---
> ### **References**
>
> * **[1]** Lin, H., Zhao, G., Zhang, O., Huang, Y., Wu, L., Tan, C., ... & Li, S. Z. CBGBench: Fill in the Blank of Protein-Molecule Complex Binding Graph. In The Thirteenth International Conference on Learning Representations.

---

### Decision · Action_Editor_iyXG · 2026-01-04

**Recommendation:** Accept as is

**Additional Comments:**

The authors made a large amount of useful comments on the work, and it is important that the authors make sure the manuscript is updated with respect to the comments and rebuttal discussion.

**Audience:**

Yes

**Audience Explanation:**

Structure-based drug design is a timely research topic in machine learning with numerous ML models being proposed for the task. The results are surely interesting for the community.

All reviewers agree.

**Claims And Evidence:**

Yes

**Claims Explanation:**

This is a benchmark paper that provides a comprehensive comparison of methods for structure-based drug design across the major modelling families. The main findings are that 3D models tend to excel at optimising binding while 1D models in molecular metrics, with 2D models being striking a balance between them. The findings are backed by empirical evidence.

3/4 reviewers agree, while one reviewer criticised the empirical setup to use outdated metrics, have some inconsistencies in the method performance, and lack of statistical analyses. The authors updated the manuscript with statistical analysis, and made convincing arguments on their metrics being sufficiently descriptive.